# What do adversarial images tell us about human vision?

Marin Dujmović[†]*, Gaurav Malhotra[†], Jeffrey S Bowers[†]

School of Psychological Science, University of Bristol, Bristol, United Kingdom

**Abstract** Deep convolutional neural networks (DCNNs) are frequently described as the best current models of human and primate vision. An obvious challenge to this claim is the existence of *adversarial images* that fool DCNNs but are uninterpretable to humans. However, recent research has suggested that there may be similarities in how humans and DCNNs interpret these seemingly nonsense images. We reanalysed data from a high-profile paper and conducted five experiments controlling for different ways in which these images can be generated and selected. We show human-DCNN agreement is much weaker and more variable than previously reported, and that the weak agreement is contingent on the choice of adversarial images and the design of the experiment. Indeed, we find there are well-known methods of generating images for which humans show no agreement with DCNNs. We conclude that adversarial images still pose a challenge to theorists using DCNNs as models of human vision.

## Introduction

Deep convolutional neural networks (DCNNs) have reached, and in some cases exceeded, human performance in many image classification benchmarks such as `ImageNet` (*He et al., 2015*). In addition to having obvious commercial implications, these successes raise questions as to whether DCNNs identify objects in a similar way to the inferotemporal cortex (IT) that supports object recognition in humans and primates. If so, these models may provide important new insights into the underlying computations performed in IT. Consistent with this possibility, a number of researchers have highlighted various functional similarities between DCNNs and human vision (*Peterson et al., 2018*) as well as similarities in patterns of activation of neurons in IT and units in DCNNs (*Yamins and DiCarlo, 2016*). This has led some authors to make strong claims regarding the theoretical significance of DCNNs to neuroscience and psychology. For example, *Kubilius et al., 2018* write: 'Deep artificial neural networks with spatially repeated processing (a.k.a., deep convolutional [Artificial Neural Networks]) have been established as the best class of candidate models of visual processing in primate ventral visual processing stream' (p.1).

One obvious problem in making this link is the existence of *adversarial* images. These are 'inputs to machine learning models that an attacker has intentionally designed to cause the model to make a mistake' (*Goodfellow et al., 2017*). *Figure 1* shows examples of two types of adversarial images. On first impression, it seems inconceivable that these adversarial images would ever confuse humans. There is now a small industry of researchers creating adversarial attacks that produce images which DCNNs classify in bizarre ways (*Akhtar and Mian, 2018*). The confident classification of these adversarial images by DCNNs suggests that humans and current architectures of DCNNs perform image classification in fundamentally different ways. If this is the case, the existence of adversarial images poses a challenge to research that considers DCNNs as models of human behaviour (e.g., *Kubilius et al., 2018*; *Ritter et al., 2017*; *Peterson et al., 2017*; *Cichy and Kaiser, 2019*; *Dodge and Karam, 2017*), or as plausible models of neural firing patterns in primate and human visual cortex (e.g., *Khaligh-Razavi and Kriegeskorte, 2014*; *Cadieu et al., 2014*; *Rajalingham et al., 2018*; *Yamins et al., 2014*; *Eickenberg et al., 2017*; *Cichy et al., 2016*).

*For correspondence:
marin.dujmovic@bristol.ac.uk

[†]These authors contributed equally to this work

Competing interests: The authors declare that no competing interests exist.

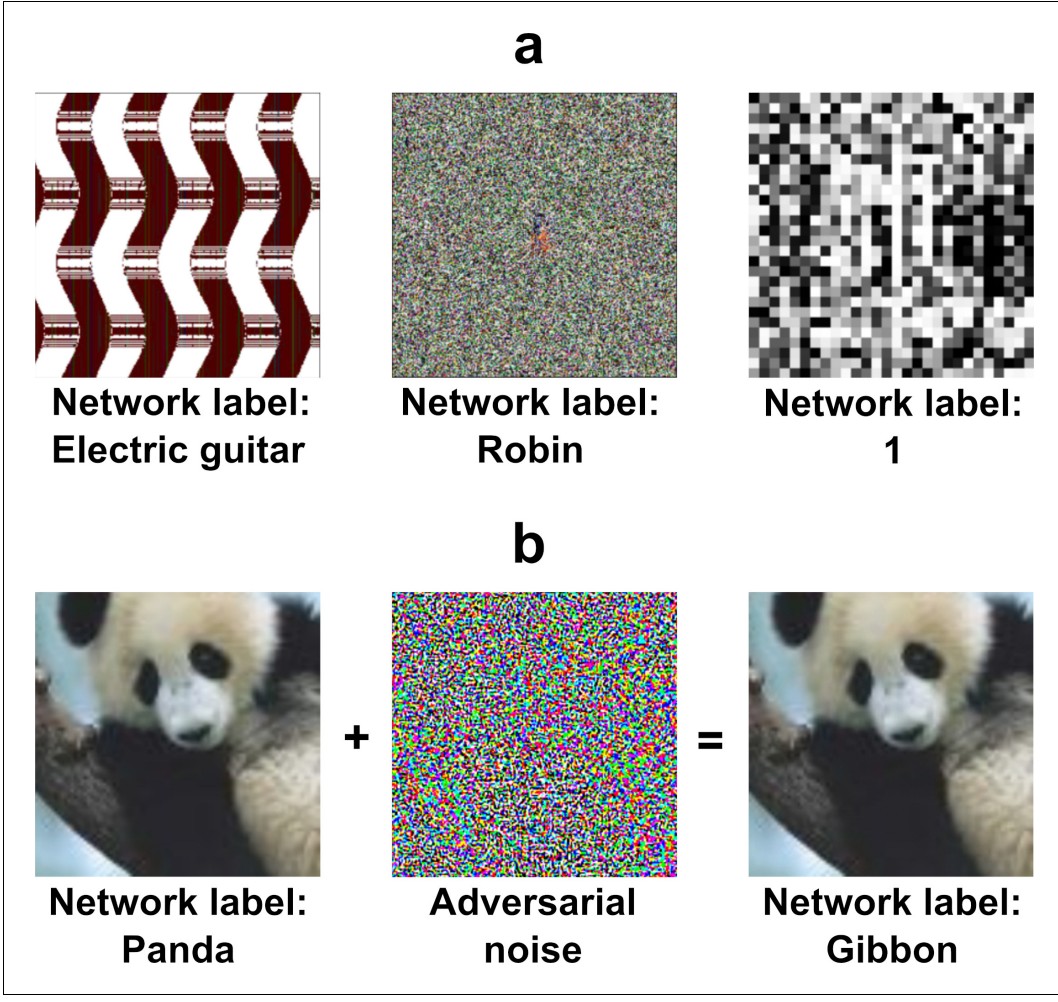

**Figure 1.** Examples of two types of adversarial images. (a) *fooling adversarial images* taken from *Nguyen et al., 2015* that do not look like any familiar object. The two images on the left (labelled 'Electric guitar' and 'Robin') have been generated using evolutionary algorithms using *indirect* and *direct* encoding, respectively, and classified confidently by a DCNN trained on `ImageNet`. The image on the right (labelled '1') is also generated using an evolutionary algorithm using *direct* encoding and it is classified confidently by a DCNN trained on `MNIST`. (b) An example of a *naturalistic adversarial image* taken from *Goodfellow et al., 2014* that is generated by perturbing a naturalistic image on the left (classified as 'Panda') with a high-frequency noise mask (middle) and confidently (mis) classified by a DCNN (as a 'Gibbon').

However, some recent studies have suggested that there may, in fact, be theoretically relevant overlap between DCNNs and humans in how they process these adversarial images. *Zhou and Firestone, 2019* (*Z&F* from here on) recently reported that humans can reliably *decipher* adversarial images that, on first viewing, look uninterpretable (as in *Figure 1a*). The authors took a range of published adversarial images that were claimed to be uninterpretable by humans and, in a series of experiments, they showed those images to human subjects next to the DCNN's preferred label and various foil labels. They reported that, over the course of an experimental session, a high percentage of participants (often close to 90%) chose the DCNN's preferred label at above-chance rates. Furthermore, they report evidence that humans appreciate subtler distinctions made by the machine rather than simply agreeing on the basis of some superficial features (such as predicting 'bagel' rather than 'pinwheel' when confronted with an image of a round and yellow blob). These results are important because they speak to an important theoretical question that *Z&F* pose in the first line of their abstract: 'Does the human mind resemble the machine-learning systems that mirror

its performance?' (p.1). The high level of agreement they report seems to suggest the answer is 'yes'.

Here we show that the agreement between humans and DCNNs on adversarial images was weak and highly variable between participants and images. The remaining agreement appeared to reflect participants making educated guesses based on some superficial features (such as colour) within images and the limited response alternatives presented to them. We then carried out five experiments in which we systematically manipulated factors that can contribute to an observed agreement between humans and DCNNs in order to better understand how humans interpret adversarial images. The experiments demonstrate that the overlap between human and DCNN classification is contingent upon various details of the experimental design such as the selection of adversarial images used as stimuli, the response alternatives presented to participants during the experiment, the adversarial algorithm used to generate the images and the dataset on which the model was trained. When we controlled for these factors, we observed that the agreement between humans and DCNNs dropped to near chance levels. Even when adversarial images were selected such that multiple DCNNs confidently assigned the same label to these images, humans seldom agreed with the machine label, especially when they had to choose between response alternatives that contained superficial features present within these images. We also show that it is straightforward to generate adversarial images that fool networks trained on `ImageNet` but are truly meaningless to human participants, irrespective of how the stimuli are selected or response alternatives are presented to a participant. We take the findings to highlight a dramatic difference between human and DCNN object recognition.

## Results

### Reassessing the level of agreement in *Zhou and Firestone, 2019*

Our first step, in trying to understand the agreement between humans and DCNNs observed by *Z&F*, was to assess how well their methods reflect the degree of agreement between humans and DCNNs. *Z&F* conducted seven experiments in which they measured agreement by computing the number of trials on which the participants matched the DCNN's classification and working out whether this number is numerically above or below chance level. So in Experiment 3, for example, a participant is shown an adversarial image on each trial and asked to choose one amongst 48 labels for that image. Each trial was independent, so a participant can choose any of the 48 labels for each image. Chance level is 1/48, so if the participant chooses the same label as the DCNN on two or more trials, they were labelled as agreeing with the DCNN. In addition, half of the participants who agreed with the DCNN on only 1/48 trials were also counted towards the number of participants who agreed with the DCNN. When computed in this manner, *Z&F* calculated that 142 out of 161 (88%) participants in Experiment 3a, and 156 out of 174 (90%) in Experiment 3b agreed with the DCNN at above chance levels.

This is a reasonable way of measuring agreement if the goal is to determine whether agreement between humans and DCNNs is statistically above chance levels. However, if the goal is to measure the *degree of agreement*, this method may be misleading and liable to misinterpretation. Firstly, the rates of agreement obtained using this method ignore inter-individual variability and assign the same importance to a participant that agrees on 2 out of 48 trials as a participant who agrees on all 48 trials with the DCNN. Secondly, this method obscures information about the number of trials on which humans and DCNNs disagree. So even if every participant disagreed with the network on 46 out of 48 trials, the rate of agreement, computed in this manner, would be 100% and even a sample of blindfolded participants would show 45% agreement (see Materials and methods). In fact, not a single participant in Experiments 3a and 3b (from a total of 335 participants) agreed with the model on a majority (24 or more) of trials, yet the level of agreement computed using this method is nearly 90%.

A better way of measuring the degree of agreement is to simply report the average agreement. This can be calculated as the mean percentage of images (across participants) on which participants and DCNNs agree. This method overcomes the disadvantages mentioned above: it takes into consideration the level of agreement of each participant (a participant who agrees on 4/48 trials is not treated equivalently to a participant who agrees on 48/48 trials), and it reflects both the levels of

agreement and disagreement observed (so a mean agreement of 100% would indeed mean that participants agreed with the DCNN classification on all the trials). *Z&F* reported mean agreement for the first of their seven experiments and in *Table 1* we report mean agreement levels in all their experiments. Viewed in this manner, it is clear that the degree of agreement in the experiments carried out by *Z&F* is, in fact, fairly modest and far from 'surprisingly universal' (p.2) or 'general agreement' (p.4) the authors reported.

## Reassessing the basis of the agreement in *Zhou and Firestone, 2019*

Although the mean agreement highlights a much more modest degree of agreement, it is still the case that the agreement was above chance. Perhaps the most striking result is in *Z&F*'s Experiment 3 where participants had to choose between 48 response alternatives and mean agreement was ~10% with chance being ~2%. Does this consistent, above chance agreement indicate that there are common underlying principles in the way humans and DCNNs perform object classification?

In order to clarify the basis of overall agreement we first assessed the level of agreement for each of the 48 images separately. As shown in *Appendix 1—figure 1*, the distribution of agreement levels was highly skewed and had a large variance. There was a small subset of images that looked like the target class (such as the Chainlink Fence, which can be seen in *Appendix 1—figure 2*) and participants showed a high level of agreement with DCNNs on these images. Another subset of images with lower (but statistically significant) levels of agreement contained some features consistent with the target class, such as the Computer Keyboard which contains repeating rectangles. But agreement on many images (21/48) was at or below chance levels. This indicates that the agreement is largely driven by a subset of adversarial images, some of which (such as the Chainlink Fence) simply depict the target class.

We also observed that there was only a small subset of images on which participants showed a clear preference amongst response alternatives that matched the DCNN's label. For most adversarial images, the distribution of participant responses across response alternatives was fairly flat (see *Appendix 1—figures 4–6*) and the most frequent human response did not match the machine label even when agreement between humans and DCNNs was above chance (see *Appendix 1—figure 2*). In fact, the label assigned to the image by DCNNs was ranked 9th (Experiment 3a) or 10th (Experiment 3b) on average. 75% of the adversarial images in Experiment 3a and 79.2% in Experiment 3b were not assigned the label chosen by the DCNN with highest frequency (*Appendix 1—figures 2* and *3*). This indicates that most adversarial images do not contain features required by humans to uniquely identify an object category.

Collectively, these findings suggest that the above chance level of agreement was driven by two subsets of images. A very small subset of images have features that humans can perceive and are

**Table 1.** Mean DCNN-participant agreement in the experiments conducted by *Zhou and Firestone, 2019*

| Exp. | Test type | Mean agreement | Chance |
|---|---|---|---|
| 1 | Fooling 2AFC [N15] | 74.18% (35.61/48 images) | 50% |
| 2 | Fooling 2AFC [N15] | 61.59% (29.56/48 images) | 50% |
| 3a | Fooling 48AFC [N15] | 10.12% (4.86/48 images) | 2.08% |
| 3b | Fooling 48AFC [N15] | 9.96% (4.78/48 images) | 2.08% |
| 4 | TV-static 8AFC [N15] | 28.97% (2.32/8 images) | 12.5% |
| 5 | Digits 9AFC [P16] | 16% (1.44/9 images) | 11.11% |
| 6 | Naturalistic 2AFC [K18] | 73.49% (7.3/10 images) | 50% |
| 7 | 3D Objects 2AFC [A17] | 59.55% (31.56/53 images) | 50% |

\* To give the readers a sense of the levels of agreement observed in these experiments, we have also computed the average number of images in each experiment where humans and DCNNs agree as well as the level of agreement expected if participants were responding at chance.

[†] Stimuli sources: N15 - *Nguyen et al., 2015*; P16 - *Papernot et al., 2016*; K18 - *Karmon et al., 2018*; A17 - *Athalye et al., 2017*.

highly predictive of the target category (e.g., Chainlink Fence image that no one would call 'uninterpretable'), and another subset of images that include visible features consistent with the target category as well as a number of other categories. These category-general features (such as colour or curvature) are what *Z&F* called 'superficial commonalities' between images (*Zhou and Firestone, 2019*, p. 2). For this subset of images, the most frequent response chosen by participants does *not* usually match the label assigned by the DCNN. Participants in these cases seem to be making educated guesses using superficial features of the target images to hedge their bets. For the rest of the images agreement is at or below chance levels.

In order to more directly test how humans interpret adversarial images we carried out five experiments. First, if participants are making educated guesses based on superficial features, then agreement levels should decrease when presented with response alternatives that do not support this strategy. We test this in Experiment 1. Second, if a DCNN develops human-like representations for a subset of categories (e.g., the Chainlink Fence category for which human-DCNN agreement was high for a specific adversarial image of a chainlink fence), then it should not matter which adversarial image from these categories is used to evaluate agreement. We test this in Experiment 2. Third, if DCNNs are processing images in very different ways to humans, then it should be possible to find situations in which overall agreement levels are at absolute chance levels. In Experiment 3 we show that one class of adversarial images for the MNIST dataset generated overall chance level agreement. In Experiment 4 we show that it is straightforward to generate adversarial images for the ImageNet dataset that produce overall chance level agreement. Finally, in Experiment 5 we show that agreement levels between humans and DCNNs remain low and variable even for images that fool an ensemble of DCNNs. The findings further undermine any claim that DCCNs and humans categorize adversarial images in a similar way.

## Experiment 1: Response alternatives

One critical difference between decisions made by DCNNs and human participants in an experiment is the number of response alternatives available. For example, DCNNs trained on ImageNet will choose a response from amongst 1000 alternatives while participants will usually choose from a much smaller cohort. In Experiment 1, we tested whether agreement levels are contingent on how these response alternatives are chosen during an experiment. We chose a subset of ten images from the 48 that were used by *Z&F* and identified four *competitive* response alternatives (from amongst the 1000 categories in ImageNet) for each of these images. One of these alternatives was always the category picked by the DCNN and the remaining three were subjectively established as categories which share some superficial visual features with the target adversarial image. For example, one of the adversarial images contains a *florescent orange curve* and is confidently classified by the DCNN as a Volcano. For this image, we chose the set of response alternatives {Lighter, Missile, Table lamp, Volcano}, all of which also contain this superficial visual feature. See *Appendix 2—figure 1* for the complete list of images and response alternatives. Participants were then shown each of these ten images and asked to choose one amongst these four competitive response alternatives. Note that if humans possess a 'machine-theory-of-mind', it should not matter how one samples response alternatives as a DCNN classifies the fooling adversarial images with high confidence (>99%) in the presence of *all* 999 alternative labels, including the competing alternatives we have selected. In the control condition an independent sample of participants completed the same task but the alternative labels were chosen at *random* from the 48 used by *Z&F*.

We observed that agreement levels fell nearly to chance in the competitive condition while being well above chance in the random condition (see *Figure 2*). The mean agreement level in the competitive condition was at 28.5% (SD = 11.67) with chance being at 25%. A single sample t-test comparing the mean agreement level to the fixed value of 25% did show the difference was significant ($t(99) = 3.00, p = .0034, d = 0.30$). However, in the random condition mean agreement was 49.8% (SD = 16.02) which was both significantly above chance ($t(99) = 15.48, p<.0001, d = 1.54$) and well above agreement in the competitive condition ($t(198) = 10.75, p<.0001, d = 1.52$). Both conditions are in stark contrast to the DCNN which classified these images with a confidence >99% even in the presence of these competing categories.

These results highlight a key contrast between human and DCNN image classification. While the features in each of these adversarial images are sufficient for a DCNN to uniquely identify one amongst a 1000 categories, for humans they are not. Instead features within these images only allow

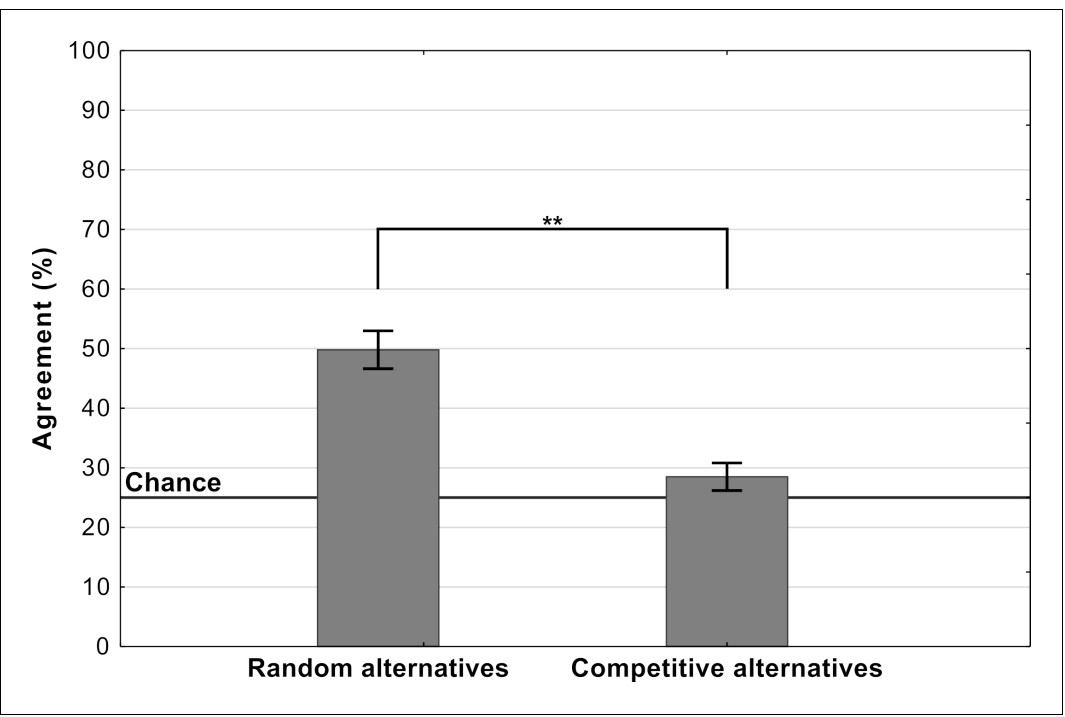

**Figure 2.** Average levels of agreement in Experiment 1 (error bars denote 95% confidence intervals).

them to identify a cohort of categories. Thus, the observed decrease in agreement between the random and competitive conditions supports the hypothesis that participants are making plausible guesses in these experiments, using superficial features (shared amongst a cohort of categories) to eliminate response alternatives.

It should be noted that *Z&F* were themselves concerned about how the choice of response alternatives may have influenced their results. Therefore, they carried out another experiment where, instead of choosing between the DCNN's preferred label and another randomly selected label, the participants had to choose between the DCNN's 1st and 2nd-ranked labels. The problem with this approach is that the DCNN generally has a very high level of confidence (>99%) in it's 1st choice. Accordingly, it is not at all clear that the 2nd most confident choice made by the network provides the most challenging response alternative for humans. The results from Experiment 1 show that when the competing alternative is selected using a different criterion, the agreement between participants and DCNNs does indeed drop to near-chance levels.

## Experiment 2: Target adversarial images

Our reanalysis above also showed that there was large variability in agreement between images. One possible explanation for this is that the DCNN learns to represent some categories (such as Chainlink Fence or Computer Keyboard) in a manner that closely relates to human object recognition while representations for other categories diverge. If there was meaningful overlap between human and DCNN representations for a category, we would expect participants to show a similar level of agreement on all adversarial images for this category as all adversarial images will capture these common features. So replacing an adversarial image from these categories with another image generated in the same manner should lead to little change in agreement. In Experiment 2 we directly tested this hypothesis by sampling two different images (amongst the five images for each category generated by *Nguyen et al., 2015*) for the same ten categories from Experiment 1. We chose the best and worst representative stimuli for each of the categories by running a pre-study (see the Materials and methods section) and labelled the two conditions as *best-case* and *worst-case*. An example of each type of image is shown in *Figure 3*.

*Figure 4* shows the mean agreement for participants viewing the *best-case* and *worst-case* adversarial images. The difference in agreement between the two conditions was highly significant

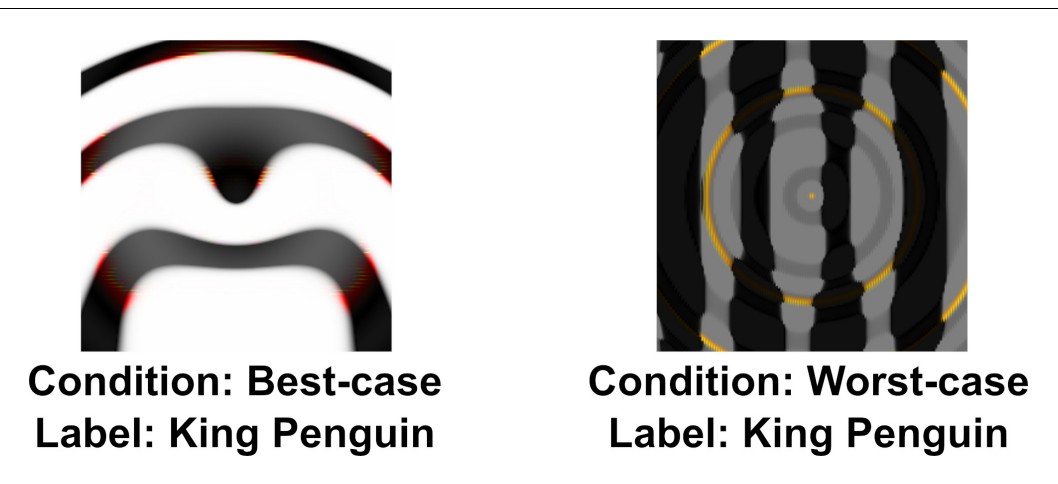

**Figure 3.** Example of *best-case* and *worst-case* images for the same category ('penguin') used in Experiment 2.

($t(198) = 22.28, p{<}.0001, d = 3.15$). Both groups showed agreement levels significantly different from chance (which was at 25%). The best-case group was significantly above chance ($t(99) = 20.12, p{<}.0001, d = 2.01$) while the worst-case was significantly below chance ($t(99) = 10.58, p{<}.0001, d = 0.99$).

Thus, we observed a large drop in agreement when we replaced one set of adversarial images with a different set, and there was no evidence for consistent above-chance agreement for all adversarial images from a subset of categories (see *Appendix 2—figure 2* for an item-wise breakdown). In other words, we did not observe any support for the hypothesis that DCNNs learn to represent even a subset of categories in a manner that closely relates to human object recognition.

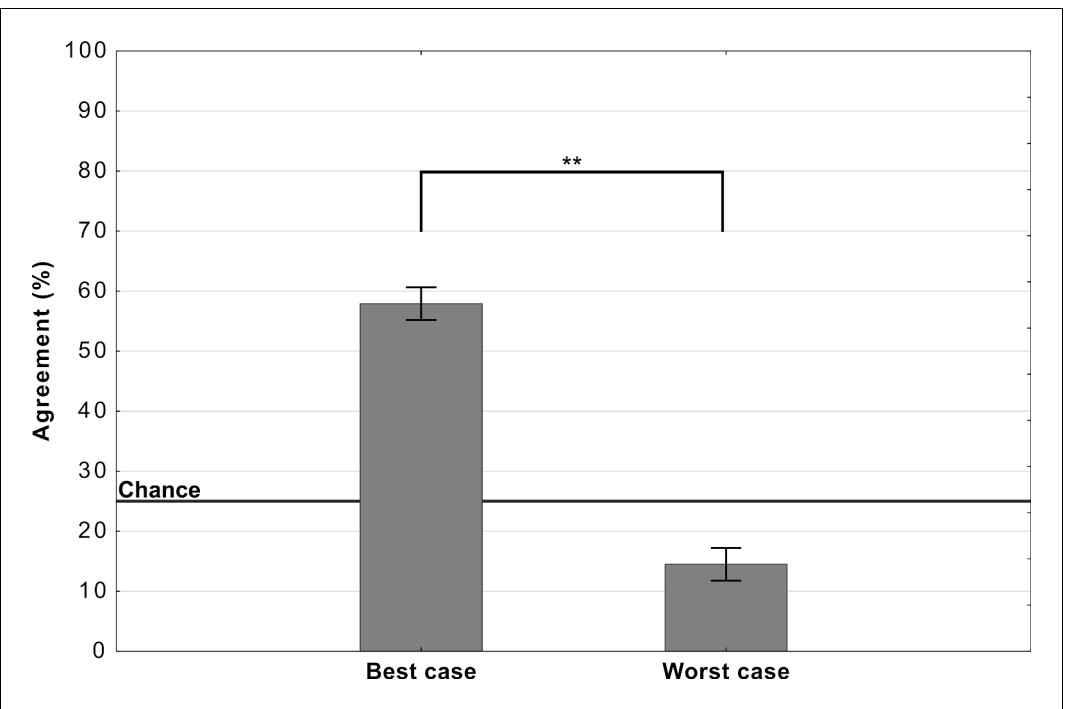

**Figure 4.** Average levels of agreement in Experiment 2 (error bars denote 95% confidence intervals).

## Experiment 3: Different types of adversarial images

Although we can easily reduce DCNN-human agreement to chance by judicially selecting the targets and foils, it remains the case that a random selection of targets and foils has led to above chance performance on this set of images. In the next experiment, we asked whether this effect is robust across different types of adversarial images. All the images in the experiments above were generated to fool a network that had been trained on `ImageNet` and belonged to the subclass of *regular* adversarial images generated by *Nguyen et al., 2015* using an *indirect encoding* evolutionary algorithm. In fact, *Nguyen et al., 2015* generated four different types of adversarial images by manipulating the type of encoding – *direct* or *indirect* – and the type of database the network was trained on – `ImageNet` or `MNIST` (see *Figure 5*). We noticed that *Z&F* used images designed to fool DCNNs trained on images from `ImageNet`, but did not consider the adversarial images designed to fool a network trained on `MNIST` dataset. To our eyes, these `MNIST` adversarial images looked completely uninterpretable and we wanted to test whether the above chance agreement was contingent on which set of images were used in the experiments.

Accordingly, we designed a 2 × 2 experiment in which we tested participants on all four conditions corresponding to the four types of images (*Figure 5*). Since `MNIST` has ten response categories and we wanted to compare results for the `MNIST` images with `ImageNet` images, we used the same 10 categories from Experiments 1 and 2 for the two `ImageNet` conditions. On each trial, participants were shown an adversarial image and asked to choose one out of ten response alternatives that remained fixed for all trials.

Mean agreement levels in this experiment are shown in *Figure 6*. We observed a large difference in agreement levels depending on the types of adversarial images. Results of a two-way repeated measures ANOVA revealed a significant effect of dataset on agreement levels ($F(1,197) = 298.62, p<.0001, \eta_p^2 = 0.60$). Participants agreed with DCNN classification for images designed to fool `ImageNet` classifiers significantly more than for images designed to fool `MNIST` classifiers. Participants also showed significantly larger agreement for indirectly-encoded compared to directly-encoded images ($F(1,197) = 67.57, p<.0001, \eta_p^2 = 0.26$). The most striking observation was that agreement dropped from 26% for `ImageNet` images to near chance for `MNIST` images. Participants were slightly above chance for indirectly-encoded `MNIST` images ($t(197)>6.30, p<.0001, d = 0.44$) and at chance agreement for directly-encoded `MNIST` images ($t(197) = 1.03, p = 0.31$).

In addition to the between-condition differences, we also found high within-condition variability for the `ImageNet` images. We observed that this was because agreement was driven by a subset of adversarial images (see *Appendix 2—figure 3* for a break down). Thus, even for these `ImageNet` images, DCNN representations do not consistently overlap with representations used by humans.

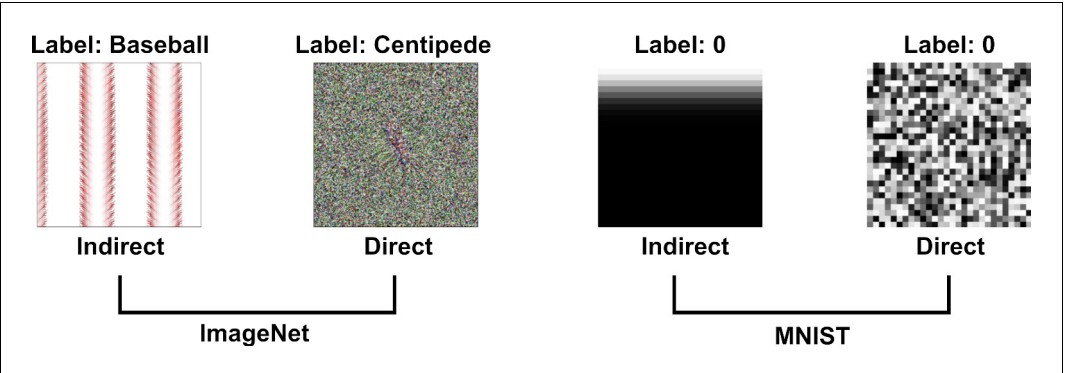

**Figure 5.** Examples of images from *Nguyen et al., 2015* used in the four experimental conditions in Experiment 3. Images are generated using an evolutionary algorithm either using the *direct* or *indirect* encoding and generated to fool a network trained on either `ImageNet` or `MNIST`.

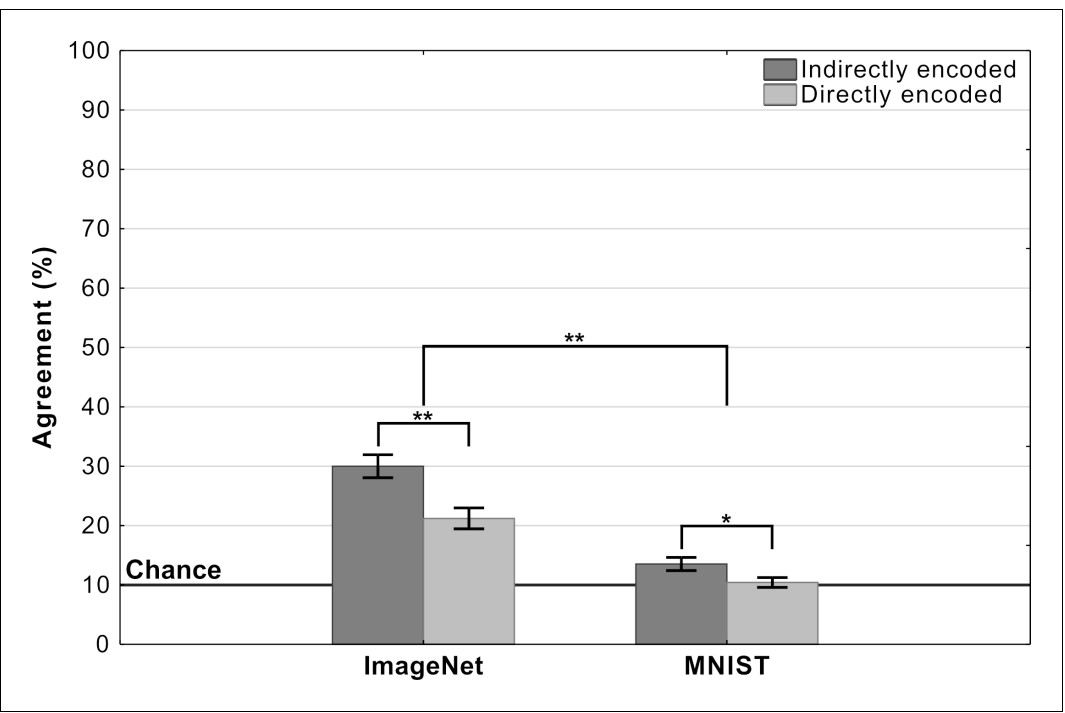

**Figure 6.** Agreement (mean percentage of images on which a participant choices agree with the DCNN) as a function of experimental condition in Experiment 3 (error bars denote 95% confidence intervals).

## Experiment 4: Generating fooling images for `ImageNet`

Experiment 3 showed that it is straightforward to obtain overall chance level performance on the `MNIST` images, and this raises the obvious question of whether it is also straightforward to observe chance performance for adversarial images designed to fool `ImageNet` classifiers? In order to test this we generated our own irregular (TV-static like) adversarial images using a standard method of generating adversarial images (see Materials and methods section). Each of these images was confidently classified as one out of a 1000 categories by a network trained on `ImageNet`. Participants were presented three of these adversarial images and asked to choose the image that most closely matches the target category (see inset in *Figure 7*). In half of the trials participants were shown adversarial images that were generated to fool AlexNet while in the other half they were shown adversarial images generated to fool Resnet-18.

Results of the experiment are shown in *Figure 7*. For both types of images, the agreement between participants and DCNNs was at chance. Additionally, we ran binomial tests for each image in order to determine whether the number of participants which agreed with DCNN classification was significantly above chance and the results showed that not a single image showed agreement that was significantly above chance. Clearly, participants could not find meaningful features in any of these images, while networks were able to confidently classify each of these images.

## Experiment 5: Transferable adversarial images

In the experiments above we observe that while DCNNs are vulnerable to adversarial attacks (they classify these images with extremely high confidence), participants do not show such a vulnerability or even a consistent agreement with the DCNN classification. But it does *not* necessarily follow that DCNNs are poor models of biological vision. In fact, there are many different methods of generating adversarial images (*Akhtar and Mian, 2018*) and some do not transfer even from one DCNN to another, and this does not merit the conclusion that the different the DCNNs function in fundamentally different ways (indeed, current DCNNs are highly similar to one another, by design). In a similar manner, the fact that adversarial images do not transfer between DCNNs and humans does not, by itself, support the conclusion that the human visual system and DCNNs are fundamentally different.

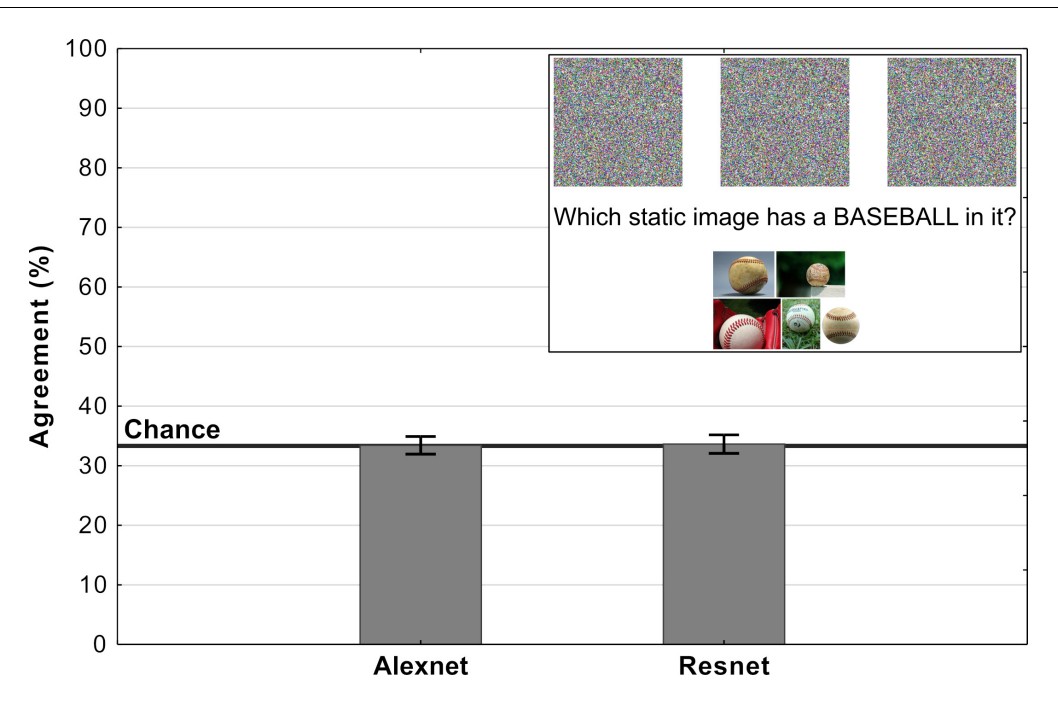

**Figure 7.** Average levels of agreement in Experiment 4 (error bars denote 95% confidence intervals). The inset depicts a single trial in which participants were shown three fooling adversarial images and naturalistic examples from the target category. Their task was to choose the adversarial image which contained an object from the target category.

In order to provide an stronger test of the similarity of DCNNs and human vision, we asked whether adversarial images that fool multiple DCNNs are decipherable by humans. If indeed there are some underlying and reliable similarities in how stimuli are processed in DCNNs and humans, then it might be expected that highly transferable DCNN adversarial attacks should also lead to higher human to network agreement.

So in the next experiment, we chose 20 adversarial images that 10 DCNNs classify with high confidence and high between-network agreement (see the Materials and methods section for details). The experiment then follows the same procedure as Experiment 1, where a participant is shown an adversarial image on each trial and asked to choose a label from four response alternatives. Like Experiment 1, participants are assigned to one of two conditions. In the *random alternatives* condition, participants were shown the network label and three other labels, which were randomly drawn from the remaining 19 labels. In the *competitive alternatives* condition, participants again had to choose from the network label and three alternative labels. However, in this condition the labels were chosen amongst the 999 remaining category labels in `ImageNet` such that they contain some superficial features contained within these images (see Materials and methods for details). Note that all DCNNs classified these images with high confidence and with all 1000 `ImageNet` labels present as alternatives.

Results are depicted in *Figure 8(b)*. There was a significant difference between the two conditions $t(198) = 16.37, p<.0001, d = 2.32$. Additionally, both conditions were significantly different from chance. Agreement in the *random alternatives* was above chance ($t(99) = 18.66, p<.0001, d = 1.87$) and below chance in the *competitive labels* condition ($t(99) = 3.13, p<.01, d = 0.31$).

Thus we find very similar levels of agreement for these adversarial images, which fool multiple DCNNs, to the adversarial images from Experiment 1 (compare *Figure 8(b)* and *Figure 2*). To further examine how the DCNN-to-DCNN agreement compares to DCNN-to-human agreement, we computed the probability that two randomly sampled networks will agree on an image's label and compared it to the probability that a randomly sampled network will agree with a randomly sampled participant (see Materials and methods for details). *Figure 8(c)* shows these probabilities for the

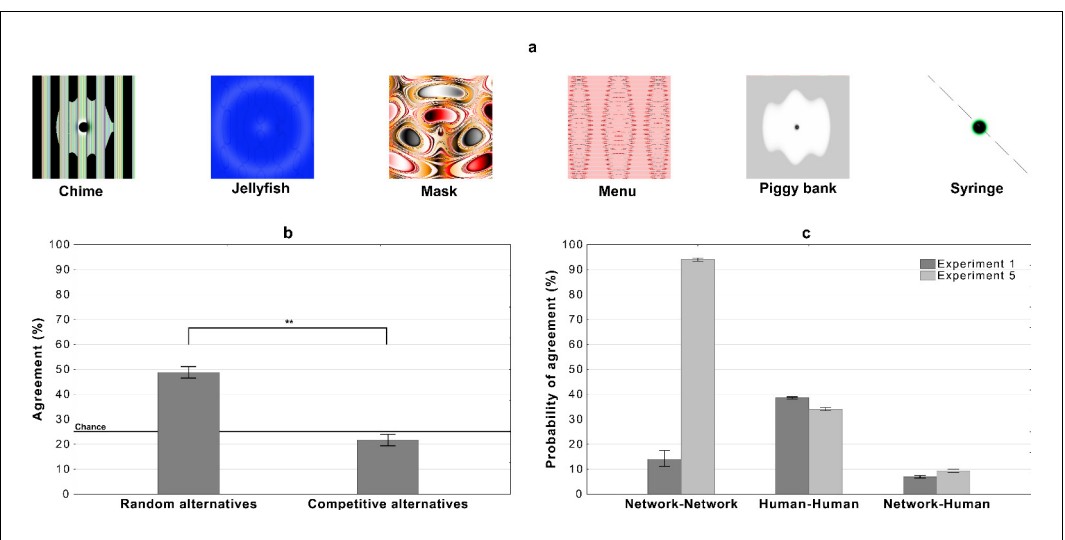

**Figure 8.** Results for images that are confidently classified with high network-to-network agreement on `Alexnet`, `Densenet-161`, `GoogLeNet`, `MNASNet 1.0`, `MobileNet v2`, `Resnet 18`, `Resnet 50`, `Shufflenet v2`, `Squeezenet 1.0`, and `VGG-16`. (**a**) Examples of images used in the experiment - for all the stimuli see *Appendix 2—figures 4* and *5*, (**b**) average levels of agreement between participants and DCNNs under the *random* and *competitive* alternatives conditions in Experiment 5, and (**c**) probability of network, human, and network to human agreement in the *competitive alternatives* condition of Experiment 1 and Experiment 5 (error bars denote 95% confidence intervals).

competitive condition for both Experiment 1 and Experiment 5. We observed that: (i) even when the probability that two networks agree on an adversarial image is larger than 90% the probability of network-human agreement is low (~10%), and (ii) the increase in probability of network-network agreement (between Experiment 1 and Experiment 5) has very little impact on human classification as the probability of human-human and network-human agreement remains much the same in the two experiments. Thus, participants showed very little agreement with DCNNs even when DCNNs agreed with each other. Interestingly, humans showed more agreement amongst themselves, consistent with the hypothesis that participants represent these adversarial images in similar ways, even though we find no evidence that these representations overlap those of the networks. This again suggests that humans and current DCNNs process these images in fundamentally different ways.

## Discussion

*Zhou and Firestone, 2019*'s claim that humans can robustly decipher adversarial images suggests that there are important similarities in how humans and DCNNs process these images, and objects more generally. However, when we examined their results using an alternative analysis, we found that the level of agreement was rather low, highly variable, and largely driven by a subset of images where participants could eliminate response alternatives based on superficial features present within these images. This was confirmed in a series of experiments that found that agreement between humans and DCNNs was contingent on the adversarial images chosen as stimuli (Experiments 2 and 3) and the response alternative presented to participants (Experiment 1). We also show that there are well-known methods for generating adversarial images that lead to overall chance level DCNN-human agreement (Experiments 3 and 4), again demonstrating that DCNNs confidently identify images on the basis of features that humans completely ignore. Furthermore, even when humans were presented with adversarial images that fooled at least 9 of 10 DCNNs, the level of agreement between humans and DCNNs remained low and variable (Experiments 5). Indeed, manipulating the level of agreement between DCNNs (by varying the adversarial images) had no impact on the level

of agreement between DCNNs and humans, or amongst humans, as highlighted in *Figure 8*. Taken together, these findings not only refute the claim that there is a robust and reliable similarity in processing these adversarial images, but also suggest that humans and current DCNNs categorize objects in fundamentally different ways.

A similar distinction between human and DCNN classification is made by *Ilyas et al., 2019*, who argue that current architectures of DCNNs are vulnerable to adversarial attacks due to their tendency for relying on *non-robust features* present in databases. These are features that are predictive of a category but highly sensitive to small perturbations of the image. It is this propensity for relying on non-robust features that makes it easy to generate adversarial images that are completely uninterpretable by humans but classified confidently by the network (Experiment 4). A striking example of DCNNs picking up on non-robust features was recently reported by *Malhotra et al., 2020* who showed that DCNNs trained on a CIFAR-10 dataset modified to contain a single diagnostic pixel per category, learn to categorize images based on single pixels ignoring everything else in the image. Humans, by contrast, tend to use robust features of objects, such as their shape, for classifying images (*Biederman and Ju, 1988*).

We would like to note that we are not claiming that there is no role played by superficial and non-robust features in human object recognition. In a recent study, *Elsayed et al., 2018* asked human participants to classify naturalistic adversarial images (see *Figure 1(b)*) when these images were briefly flashed (for around 70 ms) on the screen. They found that there is a small, but statistically significant, effect of the adversarial manipulation on choices made by participants (i.e., participants were slightly more likely to classify a 'cat' image as a 'dog' when the image was adversarially perturbed towards a 'dog'). Thus, these results seem to suggest that humans are sensitive to the same type of non-robust features that lead to adversarial attacks on DCNNs. However, it is important to note here that the size of these effects is small: while human accuracy drops by less than 10% when normal images are replaced by adversarially perturbed images, DCNNs (mis)classify these adversarially perturbed images with high confidence. These findings are consistent with our observation that some adversarial images capture some superficial features that can be used by participants to make classification decisions, leading to an overall above-chance agreement.

It should also be noted that we have only considered a small fraction of adversarial images here and, like Experiment 4, there are many other types of adversarial attacks that produce images that seem completely undecipherable for humans. It could be that humans find these images completely uninterpretable due to the difference in *acuity* of human and machine vision (a line taken by *Z&F*). There are two reasons why we think a difference in acuity cannot be the primary explanation of the difference between human and machine perception of adversarial images. Firstly, we have shown above that the very same algorithm produced some images that supported above chance agreement and other images that supported no agreement (for example, *Appendix 2—figure 2*). There is no reason to believe that the two sets of images are qualitatively different, with DCNNs selectively exploiting subliminal features only when overall agreement levels are chance. Secondly, a wide variety of adversarial attacks clearly do not rely on subtle visual features that are below human perceptual threshold. This includes semantic adversarial attacks that occur when the colour of an images is changed (*Hosseini and Poovendran, 2018*), or attacks that cause incorrect classification by simply change the pose of an image (*Alcorn et al., 2019*), etc. These are all dramatic examples of differences between DCNNs and humans that cannot be attributed to the acuity of human perceptual frontend. Rather they reflect the fact that current architectures of DCNNs are often relying on visual features that humans can see but ignore.

Of course, it might be possible to modify DCNNs so that they perform more like humans in our adversarial tasks. For example, training similar models on data sets that are more representative of human visual experience might reduce their susceptibility to adversarial images and lead DCNNs to produce more variable responses in our tasks as a consequence of picking up on superficial visual features. In addition, modifying the architectures of DCNNs or introducing new ones may lead to a better DCNN-human agreement on these tasks (for example, capsule networks [*Sabour et al., 2017*]). But researchers claiming that current DCNNs provide the best models of visual processing in primate ventral visual processing stream need to address this striking disconnect between the two systems.

To conclude, our findings with fooling adversarial images pose a challenge for theorists using current DCNNs trained on data sets like `ImageNet` as psychological models of human object

identification. An important goal for for future research is to develop models that are sensitive to the visual features that humans rely on, but at the same time insensitive to other features that are diagnostic of object category but irrelevant to human vision. This involves identifying objects on the basis of shape rather than texture or color or other diagnostic features (*Geirhos et al., 2018*; *Baker et al., 2018*), where vertices are the critical components of images (*Biederman, 1987*), where Gestalt principles are used to organize features (*Pomerantz and Portillo, 2011*), where relations between parts are explicitly coded (*Hummel and Stankiewicz, 1996*), where features and objects are coded independently of retinal position (*Blything et al., 2019*) size (*Biederman and Cooper, 1992*), left/right reflection (*Biederman and Cooper, 1991*), etc. When DCNNs rely on these set of features, we expect they will not be subject to adversarial attacks that seem so bizarre to humans, and will show the same set of of strengths and weakness (visual illusions) that characterize human vision.

## Materials and methods

### Reassessing agreement: Blindfolded participants

If a participant is blindfolded and chooses one of 48 options randomly on 48 trials, the probability of them making the same choice as the DCNN on $k$ trials is given by the binomial distribution $\binom{48}{k} p^k(1-p)^{48-k}$, where $p = \frac{1}{48}$. Substituting different values of $k$, one can compute that 37.2% of these blindfolded participants will agree with the DCNN on 1 trial, 18.6% will agree on 2 trials, 6% will agree on 3 trials, and so on. To compute the proportion of participants who agree with the DCNN, *Zhou and Firestone, 2019* count all participants who agree on 2 or more trials as agreeing with the DCNN (chance is 1 out of 48 trials) and half of the participants that agree on exactly 1 trial. Thus, summing up all the blindfolded participants that agree on 2 or more trials and half of those who agree on exactly 1 trial, this method will show ~45% agreement between participants and the DCNN.

### Experiment 1

This experiment examined whether agreement between humans and DCNNs depended on the response alternatives presented to participants. We tested $N = 200$ participants and each participant completed 10 trials. During each trial, participants were presented a fooling adversarial image and four response alternatives underneath the image and asked to choose one of these alternatives. Participants indicated their response by moving their cursor to the response alternative and clicking. We selected 10 fooling adversarial images from amongst the 48 images used by *Z&F* in in their Experiments 1–3. Each of these images was classified with >99% confidence by a DCNN which was trained to classify the `ImageNet` dataset. We selected these 10 adversarial images to minimise semantic and functional overlap in the labelled categories (for example, we avoided selecting both 'computer keyboard' and 'remote control'). The experiment consisted of two conditions, which differed in how the response alternatives were chosen on each trial. In the 'Competitive' condition ($N = 100$) we chose four response alternatives that subjectively seemed to contain one or more visual features that were present in the adversarial image. One of these response alternatives was always the label chosen by the DCNN. The other three were chosen from amongst 1000 `ImageNet` class labels. This was again done to minimise a semantic or functional overlap with the target class (e.g. an alternative for the 'baseball' class was *parallel bars* but not *basketball*). All ten images and the four competitive response alternatives for each image are shown in *Appendix 2—figure 1* In the 'Random' condition ($N = 100$) the three remaining alternative responses were drawn at random (on each trial) from the aforementioned 48 target classes from Experiment 3 in *Zhou and Firestone, 2019*. We randomised the order of images, as well as the order of the response alternatives for each participant.

### Experiment 2

This experiment was designed to examine whether all fooling adversarial images for a category show similar levels of agreement between humans and DCNNs. The experiment's design was the same as Experiment 1 above, except participants were now randomly assigned to the 'best-case'

($N = 100$) and 'worst-case' ($N = 100$) conditions. In each condition, participants again completed 10 trials and on each trial, they saw an adversarial image and four response alternatives. One of these alternatives was the category chosen by a DCNN with >99% confidence and the other three were randomly drawn from amongst the 48 categories used by *Z&F* in their Experiments 1–3. The difference between the 'best-case' and 'worst-case' conditions was the adversarial image that was shown to the participants on each trial.

In order to choose the best and worst representative image for each of the categories we ran a pre-study. Each images used by *Z&F* in their Experiments 1–3 was chosen from a set consisting of five adversarial images for that category generated by *Nguyen et al., 2015*. In the pre-study, participants ($N = 100$) were presented all five fooling images and asked to choose an image that was most-like and least-like a member from that category (e.g. most like a computer keyboard). Then, during the study, participants in the 'best-case' condition were shown the image from each category that was given the *most-like* label with the highest frequency. Similarly, participants in the 'worst-case' condition were shown images that were labelled as *least-like* with the highest frequency. DCNNs showed the same confidence in classifying both sets of images. We again randomised the order of presentation of images.

## Experiment 3

The experiment consisted of four experimental conditions in a 2 × 2 repeated measures design (every participant completed each condition). The first factor of variation was the database on which the DCNNs were trained – `ImageNet` or `MNIST` with one condition containing images designed to fool `ImageNet` and the other containing images designed to fool `MNIST` classifiers. The second factor of variation was the evolutionary algorithm used to generate the adversarial images – *direct* or *indirect*. The indirect encoding method leads to adversarial images which have regular features (e.g. edges) that often repeat, while the direct encoding method leads to noise-like adversarial images. All of the images were from the seminal *Nguyen et al., 2015* paper on fooling images. The `MNIST` dataset consists of ten categories (corresponding to handwritten numbers between 0 and 9), while `ImageNet` consists of 1000 categories. As we wanted to compare agreement across conditions, we selected ten images (from ten different categories) for both datasets. The indirectly-encoded `ImageNet` images were the same as the ones in Experiment 1 while the images for the other three conditions were randomly sampled from the images generated by *Nguyen et al., 2015*. Participants were shown one image at a time and asked to categorize it as one of ten categories (category labels were shown beneath the image). One of these ten categories was the label assigned to the image by a DCNN. Therefore, chance level agreement was 10%. The participants had to click on the label they thought represented what was in the image. The order of conditions was randomized for each participant and the order of images within each condition was randomized as well. A total of $N = 200$ participants completed the study. Two participants were excluded from analysis because their choices were made with average response times below 500 ms indicating random clicking rather than actually making decisions based on looking at the images themselves.

## Experiment 4

In this experiment we used the Foolbox package (*Rauber et al., 2017*) to generate images that fool DCNNs trained on `ImageNet`. The experiment consisted of two conditions, one with images designed to fool AlexNet (*Krizhevsky et al., 2012*) and the other with images designed to fool ResNet-18, both trained on `ImageNet`. We generated our own adversarial images by first generating an image in which each pixel was independently sampled and successively modifying this image using an Iterative Gradient Attack based on the fast gradient sign method (*Goodfellow et al., 2014*) until a DCNN classified this image as a target category with a >99% confidence. The single trial procedure mirrors Experiment 4 from *Z&F*. Participants ($N = 200$) were shown three of the generated images and a set of five real-world example images of the target class (see Inset in *Figure 7*). They were asked to choose the adversarial image which contained an object from the target class. The example images were randomly chosen from the `ImageNet` dataset for each class. Each participant completed both experimental conditions. The order of trials was randomized for each participant.

## Experiment 5

The experiment mirrors Experiment 1 in procedure and experimental conditions. Participants ($N = 200$) were sequentially shown 20 images and asked to choose one out of four response alternatives for each image. The order of presentation of these images was randomized. The stimuli were chosen from ten independent runs (a total of 10000 images) of the evolutionary algorithm used in *Nguyen et al., 2015* which were kindly provided to us by the first author. The images were selected such that at least 9 out of 10 networks classify the images as the same category with high confidence (median confidence of 92.61%). Before settling on the final set of 20 images, the stimuli were checked by the first author in order to exclude any which are not in fact adversarial, but rather exemplars of the category. The DCNN models were pre-trained on `ImageNet` and are a part of the model zoo of the PyTorch framework. The models are: `Alexnet, Densenet-161, GoogLeNet, MNAS-Net 1.0, MobileNet v2, Resnet 18, Resnet 50, Shufflenet v2, Squeezenet 1.0, and VGG-16`. The input images were transformed in accordance with recommendations found in PyTorch documentation: $224 \times 224$ centre crop and normalization with $mean = [0.485, 0.456, 0.406]$ and $std = [0.229, 0.224, 0.225]$ prior to classification.

The two experimental conditions mirror Experiment 1. In the *random alternatives* condition, for each image, participants ($N = 100$) chose among labels which included the network classification and three alternatives chosen at random among the remaining 19 stimuli labels. In the *competitive alternatives* condition, participants ($N = 100$) chose among labels which included the network classification and three competitive labels. To determine these competitive labels, we conducted a pre-study, where participants ($N = 20$) were asked to generate three labels for each adversarial image. These labels were then used as a guide to select the three competitive categories from `ImageNet` while ensuring that these categories did *not* semantically overlap with the target category. Participants were assigned to one of the two conditions randomly, the order of images and label positioning on the screen were randomized for each participant. Stimuli and competitive labels can be seen in *Appendix 2—figures 4* and *5*.

## Statistical analyses

All conducted statistical analyses were two-tailed with a p-value under 0.05 denoting a significant result. In Experiments 1, 2, 4, and 5 we conducted single sample t-tests to check if agreement levels were significantly above a fixed chance level (25% in Experiments 1, 2, and 5, 33.33% in Experiment 4). We additionally ran a between-subject t-test (Experiments 1, 2, and 5) and a within-subject t-test (Experiment 4) to determine whether the difference between experimental conditions was significant. We also conduct a Binomial test in Experiment 4 to determine for how many items was agreement level significantly above chance. In Experiment 3 we ran a two-way repeated measures analysis of variance. In Experiment 5 we ran a mixed two-way analysis of variance. We report effect size measures for all tests (Cohen's d for t-tests and partial eta squared for ANOVA effects). We calculate probability of network-network, human-human, and network-human agreement in the *competitive labels* condition of Experiment 1 and 5. This was done by calculating the percentage of agreements among all possible comparisons. For example, the total number of comparisons to calculate the probability of agreement between two networks, was: 20(images) *45(number of possible combinations of two networks). The number of such comparisons which resulted in agreement between two networks divided by the total number of comparisons gives the probability of two networks agreeing when classifying an adversarial image. We conducted the same calculation on data from Experiment 1, since those stimuli were not specifically chosen to be highly transferable between networks.

## Power analysis

A sample size of $N = 200$ was chosen for each experiment which mirrors *Z&F* experiments 1–6 in order to detect similar effects. This allowed us to detect an effect size as low as $d = 0.18$ at $\alpha = .05$ with 0.80 power in within-subject and $d = 0.35$ in between-subject experiments.

## Online recruitment

We conducted all four experiments online with recruitment through the Prolific platform. Each sample was recruited from a pool of registered participants which met the following criteria. Fluent English speakers living in the UK, USA, Canada or Australia of both genders between the ages of 18

and 50 with normal or corrected to normal vision and a high feedback rating on the Prolific platform (above 90). Participants were reimbursed for their time upon successful completion through the Prolific system.

## Data availability

Data, scripts, and stimuli form all our experiments are available via the Open Science Framework at https://osf.io/a2sh5/. Stimuli from evolutionary runs producing fooling images by *Nguyen et al., 2015* can be found at https://anhnguyen.me/project/fooling/.

## Acknowledgements

This research was supported by the European Research Council Grant Generalization in Mind and Machine, ID number 741134. We would like to thank Alex Doumas, Jeff Mitchell, Milton Liera Montero and Brian Sullivan for their insights and feedback.

## Additional information

### Funding

| Funder | Grant reference number | Author |
| --- | --- | --- |
| H2020 European Research Council | 741134 | Jeffrey S Bowers |

The funders had no role in study design, data collection and interpretation, or the decision to submit the work for publication.

### Author contributions

Marin Dujmović, Conceptualization, Resources, Data curation, Software, Formal analysis, Validation, Investigation, Visualization, Methodology, Writing - original draft, Writing - review and editing; Gaurav Malhotra, Conceptualization, Resources, Supervision, Investigation, Visualization, Methodology, Writing - original draft, Writing - review and editing; Jeffrey S Bowers, Conceptualization, Resources, Supervision, Funding acquisition, Investigation, Visualization, Methodology, Writing - original draft, Writing - review and editing

### Author ORCIDs

Marin Dujmović (iD) https://orcid.org/0000-0003-1523-227X

### Ethics

Human subjects: Participants were informed about the nature of the study, and their right to withdraw during the study or to withdraw their data from analysis. The participants gave consent for anonymized data to be used for research and available publicly. The project has been approved by the IRB at the University of Bristol (application ID 76741).

### Decision letter and Author response

Decision letter https://doi.org/10.7554/eLife.55978.sa1
Author response https://doi.org/10.7554/eLife.55978.sa2

## Additional files

### Supplementary files

• Transparent reporting form

## Data availability

Data, scripts and stimuli for all of the experiments are available via the Open Science Framework (https://osf.io/a2sh5).

The following dataset was generated:

| Author(s) | Year | Dataset title | Dataset URL | Database and Identifier |
|---|---|---|---|---|
| Dujmovic M, Malhotra G, Bowers JS | 2020 | What do adversarial images tell us about human vision? | https://osf.io/a2sh5 | Open Science Framework, a2sh5 |

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

## Appendix 1

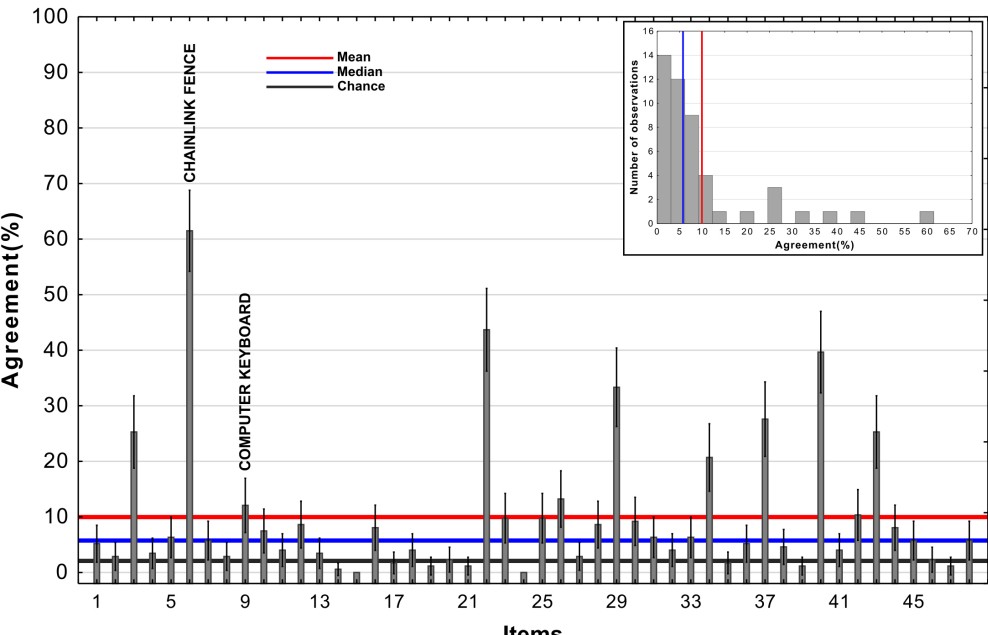

**Appendix 1—figure 1.** Agreement across adversarial images from Experiment 3b in *Zhou and Firestone, 2019*. The red line represents the mean, the blue line represents the median, and the black reference line represents chance agreement. The inset contains a histogram of agreement levels across the 48 images.

| Stimulus | DCNN label | 1 | 2 | 3 | 4 | 5 | 6 | 7 | 8 | |
|---|---|---|---|---|---|---|---|---|---|---|
| | 1. Accordion** | Traffic light 10.92% | Strawberry 8.05% | Tile roof 6.32% | **Accordion 5.17%** | Spotlight 5.17% | Electric Guitar 4.60% | Pole 4.60% | Chainlink fence 4.02% | |
| | 2. Assault rifle* | Chainlink fence 21.26% | Crossword puzzle 8.04% | Computer keyboard 6.32% | Accordion 5.17% | Soccer ball 4.60% | Ski mask 3.45% | Tile roof 3.45% | **Assault rifle 2.87%** | |
| | 3. Bagel** | **Bagel 25.28%** | Spotlight 12.07% | Traffic light 5.74% | Volcano 5.17% | Pinwheel 4.02% | Car wheel 2.87% | Chameleon 2.87% | Digital clock 2.87% | |
| | 4. Baseball* | Pole 12.64% | Four-poster bed 5.74% | Sea snake 5.74% | Chainlink fence 4.59% | Hair Clip 4.02% | Panpipe 4.02% | Roundworm 4.02% | **Baseball 3.45%** | |
| | 5. Car Wheel** | Traffic light 10.92% | Volcano 6.90% | **Car wheel 6.32%** | Obelisk 5.75% | Stethoscope 5.75% | Electric guitar 4.02% | Projector 4.02% | Digital clock 3.45% | |
| | 6. Chainlink fence** | **Chainlink fence 61.49%** | Accordion 2.87% | Hair clip 2.30% | Sea snake 2.30% | Ski mask 2.30% | Tile roof 2.30% | Computer keyboard 1.72% | Digital clock 1.72% | |
| | 7. Chameleon** | Pinwheel 6.90% | **Chameleon 5.75%** | Starfish 5.75% | Green snake 5.17% | Stethoscope 5.17% | Peacock 4.60% | Projector 4.60% | Traffic light 4.60% | |
| | 8. Comic book* | Peacock 8.05% | Projector 6.90% | Chainlink fence 5.75% | Chameleon 5.75% | Green snake 4.60% | Remote control 4.60% | Accordion 4.02% | Stethoscope 4.02% | Comic book T11 2.87% |
| | 9. Computer keyboard** | Chainlink fence 14.37% | **Computer keyboard 12.07%** | Green snake 8.05% | Tile roof 4.60% | Paddle 4.02% | Chameleon 3.45% | Digital clock 3.45% | Slot machine 3.45% | |
| | 10. Crossword puzzle** | Chainlink fence 8.62% | **Crossword puzzle 7.45%** | Tile roof 6.90% | Computer keyboard 6.32% | Accordion 4.60% | Electric guitar 4.60% | Photocopier 4.60% | Digital clock 4.02% | |
| | 11. Dial telephone* | Traffic light 9.77% | Soccer ball 8.05% | Bagel 6.32% | Car wheel 6.32% | Chameleon 6.32% | Projector 5.75% | Baseball 5.17% | Roundworm 4.60% | Dial telephone T10 4.02% |
| | 12. Digital clock** | Strawberry 14.37% | Roundworm 12.07% | **Digital clock 8.62%** | Accordion 4.60% | Remote control 4.60% | Slot machine 4.60% | Spotlight 4.02% | Computer keyboard 3.45% | |
| | 13. Electric guitar* | Sea snake 12.07% | Chainlink fence 6.32% | Hair clip 5.75% | Roundworm 5.75% | Tile roof 5.75% | Accordion 4.02% | Hand blower 4.02% | **Electric guitar 3.45%** | |
| | 14. Four-poster bed | Accordion 8.62% | Tile roof 7.47% | Freight car 5.17% | Chainlink fence 4.60% | Soccer ball 4.60% | Electric guitar 4.02% | Paddle 4.02% | Comic book 3.45% | Four poster bed T35 0.57% |
| | 15. Freight car | Projector 8.05% | Peacock 7.47% | Digital clock 6.90% | Electric guitar 6.90% | Stethoscope 6.90% | Photocopier 5.75% | Slot machine 5.17% | Chameleon 4.60% | Freight car T39 0% |
| | 16. Green snake** | **Green snake 8.05%** | Roundworm 8.05% | Spotlight 8.05% | Chameleon 6.90% | Bagel 4.02% | Digital clock 4.02% | Soccer ball 4.02% | Traffic light 4.02% | |
| | 17. Grey parrot | Bagel 14.37% | Stethoscope 8.05% | Car wheel 5.75% | Soccer ball 5.17% | Spotlight 5.17% | Vacuum 5.17% | Baseball 4.60% | Projector 4.60% | Grey parrot T15 1.72% |
| | 18. Hair clip* | Monarch butterfly 32.76% | Peacock 6.32% | Chameleon 5.75% | Ski mask 5.75% | Obelisk 4.60% | **Hair clip 4.02%** | Green snake 3.45% | Paddle 2.87% | |
| | 19. Hand blower | Computer keyboard 9.20% | Digital clock 6.90% | Stethoscope 5.17% | Volcano 4.60% | Accordion 4.02% | Hair clip 4.02% | Chameleon 3.45% | Dial telephone 3.45% | Hand blower T25 1.15% |
| | 20. King penguin* | Four-poster bed 7.47% | Car wheel 5.75% | Chainlink fence 4.60% | Stethoscope 4.60% | Projector 4.02% | Grey parrot 3.45% | Punching bag 3.45% | Slot machine 3.45% | King penguin T12 2.30% |
| | 21. Medicine chest | Slot machine 8.62% | Computer keyboard 8.05% | Chainlink fence 6.32% | Photocopier 6.32% | Electric guitar 5.17% | Panpipe 5.17% | Chameleon 4.60% | Tile roof 4.60% | Medicine chest T20 1.15% |
| | 22. Monarch butterfly** | **Monarch butterfly 43.68%** | Volcano 8.05% | Punching bag 4.02% | Hair clip 3.45% | Paddle 3.45% | Starfish 3.45% | Stethoscope 3.45% | Traffic light 3.45% | |
| | 23. Obelisk** | **Obelisk 9.77%** | Pole 6.90% | Four-poster bed 4.60% | Projector 4.60% | Tile roof 4.60% | Chainlink fence 4.02% | Crossword puzzle 4.02% | Photocopier 4.02% | |
| | 24. Paddle | Green snake 25.86% | Sea snake 6.32% | Accordion 4.02% | Obelisk 4.02% | Chainlink fence 3.45% | Chameleon 3.45% | Electric guitar 3.45% | Hand blower 3.45% | Paddle T41 0% |

* numerically above chance agreement; ** statistically above chance agreement

**Appendix 1—figure 2.** Participant responses ranked by frequency (Experiment 3b). Each row contains the adversarial image, the DCNN label for that image, the top eight participant responses. Shaded cells contain the DCNN choice, when not ranked in the top 8, it is shown at the end of the row along with the rank in brackets.

| Stimulus | DCNN label | 1 | 2 | 3 | 4 | 5 | 6 | 7 | 8 | |
|---|---|---|---|---|---|---|---|---|---|---|
| | 25. Panpipe** | Tile roof 12.07% | **Panpipe 9.77%** | Accordion 4.60% | Obelisk 4.02% | Hair clip 3.45% | Hand blower 3.45% | Pinwheel 3.45% | Pole 3.45% | |
| | 26. Peacock** | Green snake 13.22% | **Peacock 13.22%** | Sea snake 7.47% | Accordion 5.75% | Chainlink fence 5.75% | Monarch butterfly 4.02% | Chameleon 3.45% | Crossword puzzle 3.45% | |
| | 27. Photocopier* | Car wheel 16.67% | Spotlight 6.32% | Pinwheel 4.60% | Accordion 4.02% | Projector 4.02% | Traffic light 4.02% | Assault rifle 3.45% | Digital clock 3.45% | Photocopier T11 2.87% |
| | 28. Pinwheel** | **Pinwheel 8.62%** | Computer keyboard 6.90% | Projector 6.32% | Chameleon 5.75% | Peacock 5.17% | Slot machine 4.60% | Spotlight 4.60% | Monarch butterfly 4.02% | |
| | 29. Pole** | **Pole 33.33%** | Traffic light 5.17% | Comic book 4.02% | Dial telephone 4.02% | Obelisk 4.02% | Photocopier 4.02% | Chainlink fence 3.45% | Accordion 2.87% | |
| | 30. Projector** | Spotlight 19.54% | Car wheel 9.20% | **Projector 9.20%** | Stethoscope 5.75% | Soccer ball 5.17% | Traffic light 4.02% | Assault rifle 3.45% | Digital clock 3.45% | |
| | 31. Punching bag** | Pole 20.69% | **Punching bag 6.32%** | Spotlight 5.75% | Remote control 4.02% | Roundworm 4.02% | Traffic light 4.02% | Four-poster bed 3.45% | Accordion 2.87% | |
| | 32. Remote control* | Computer keyboard 10.92% | Chainlink fence 8.05% | Tile roof 5.75% | Stethoscope 5.17% | Digital clock 4.60% | Projector 4.60% | Obelisk 4.02% | **Remote control 4.02%** | |
| | 33. Roundworm** | Green snake 27.01% | **Roundworm 6.32%** | Sea snake 5.75% | Hair clip 5.75% | Electric guitar 3.45% | Digital clock 3.45% | Ski mask 3.45% | Slot machine 3.45% | |
| | 34. School bus** | **School bus 20.69%** | Accordion 5.17% | Photocopier 5.17% | Chainlink fence 4.02% | Monarch butterfly 4.02% | Medicine chest 3.45% | Tile roof 3.45% | Chameleon 2.87% | |
| | 35. Screwdriver | Accordion 12.64% | Panpipe 9.77% | Green snake 8.62% | Chainlink fence 5.75% | Computer keyboard 5.75% | Digital clock 4.02% | Slot machine 4.02% | Stethoscope 3.45% | Screwdriver T17 1.72% |
| | 36. Sea snake** | Traffic light 14.37% | Green snake 9.20% | Pole 6.90% | Spotlight 5.75% | Chameleon 5.17% | **Sea snake 5.17%** | Digital clock 4.60% | Photocopier 4.60% | |
| | 37. Ski mask** | **Ski mask 27.59%** | King penguin 8.05% | Monarch butterfly 8.05% | Peacock 4.60% | Electric guitar 3.45% | Pinwheel 3.45% | Chameleon 2.87% | Comic book 2.87% | |
| | 38. Slot machine* | Spotlight 10.35% | Pinwheel 9.20% | Crossword puzzle 5.17% | Projector 5.17% | Bagel 4.60% | **Slot machine 4.60%** | Traffic light 4.60% | Computer keyboard 4.02% | |
| | 39. Soccer ball | Chainlink fence 17.24% | Crossword puzzle 6.90% | Photocopier 6.32% | Electric guitar 5.75% | Accordion 5.17% | Traffic light 4.60% | Digital clock 2.87% | Projector 2.87% | Soccer ball T26 1.15% |
| | 40. Spotlight** | **Spotlight 39.66%** | Projector 15.52% | Traffic light 5.75% | Vacuum 4.60% | Obelisk 3.45% | Stethoscope 3.45% | Soccer ball 2.87% | Freight car 2.30% | |
| | 41. Starfish* | Sea snake 11.49% | Electric guitar 6.90% | Peacock 6.90% | Accordion 6.32% | Chainlink fence 5.17% | Slot machine 4.60% | Pinwheel 4.02% | **Starfish 4.02%** | |
| | 42. Stethoscope** | Car wheel 12.06% | **Stethoscope 10.34%** | Chainlink fence 6.90% | Spotlight 6.32% | Obelisk 4.02% | Photocopier 4.02% | Projector 4.02% | Soccer ball 4.02% | |
| | 43. Strawberry** | **Strawberry 25.29%** | Traffic light 16.67% | Slot machine 6.90% | Baseball 4.60% | Monarch butterfly 3.45% | Stethoscope 3.45% | Soccer ball 2.87% | Medicine chest 2.30% | |
| | 44. Tile roof** | Accordion 8.62% | Volcano 8.62% | **Tile roof 8.05%** | Monarch butterfly 6.90% | Chainlink fence 5.17% | Starfish 5.17% | Dial telephone 4.02% | Hand blower 4.02% | |
| | 45. Traffic light** | Projector 7.47% | Pinwheel 6.90% | Spotlight 6.32% | Car wheel 5.75% | **Traffic light 5.75%** | Stethoscope 4.02% | Obelisk 3.45% | Photocopier 3.45% | |
| | 46. Trifle* | Strawberry 9.77% | Pinwheel 7.47% | Roundworm 6.90% | Traffic light 6.90% | Spotlight 6.32% | Volcano 5.75% | Car wheel 4.60% | Stethoscope 4.60% | Trifle T13 2.30% |
| | 47. Vacuum | Green snake 4.60% | Chameleon 4.02% | Monarch butterfly 4.02% | Obelisk 4.02% | Peacock 4.02% | Pinwheel 4.02% | Projector 4.02% | Starfish 4.02% | Vacuum T31 1.15% |
| | 48. Volcano** | Pole 9.77% | Hair clip 6.90% | Spotlight 6.90% | Traffic light 6.32% | **Volcano 5.75%** | Roundworm 5.17% | Panpipe 4.60% | Punching bag 4.60% | |

* numerically above chance agreement; ** statistically above chance agreement

**Appendix 1—figure 3.** Participant responses ranked by frequency (Experiment 3b). Continued.

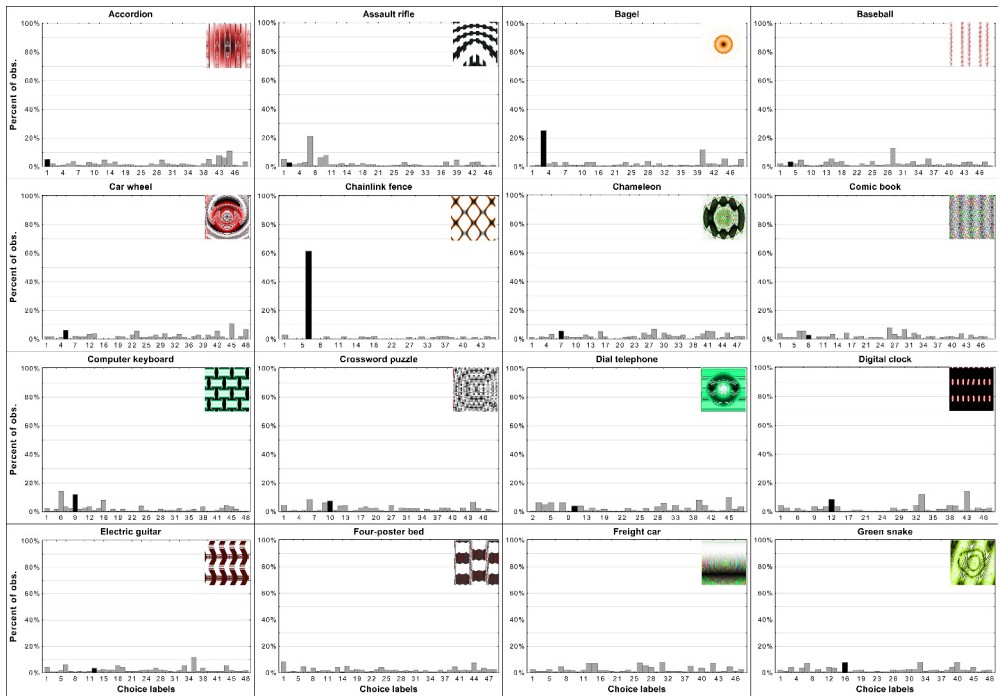

**Appendix 1—figure 4.** Per-item histograms of response choices from Experiment 3b in *Zhou and Firestone, 2019*. Each histogram contains the adversarial stimuli and shows the percentage of responses per each choice (y-axis). The choice labels (x-axis) are ordered the same way as in *Appendix 1—figures 2* and *3* from 1 to 48. Black bars indicate the DCNN choice for a particular adversarial image.

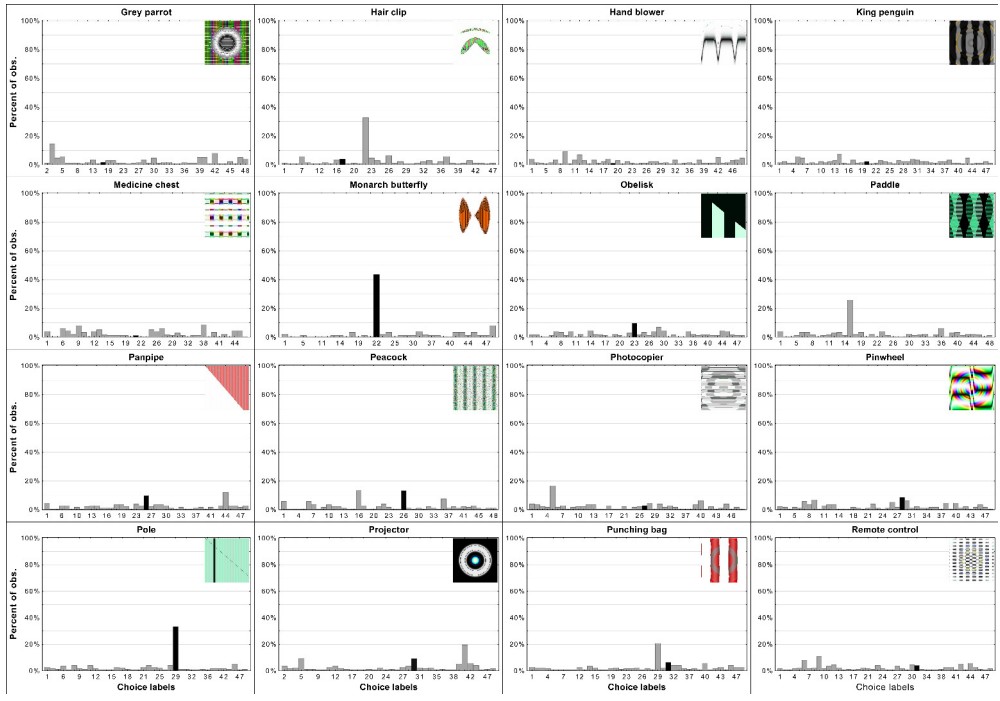

**Appendix 1—figure 5.** Per-item histograms of response choices from Experiment 3b in *Zhou and Firestone, 2019*. Continued.

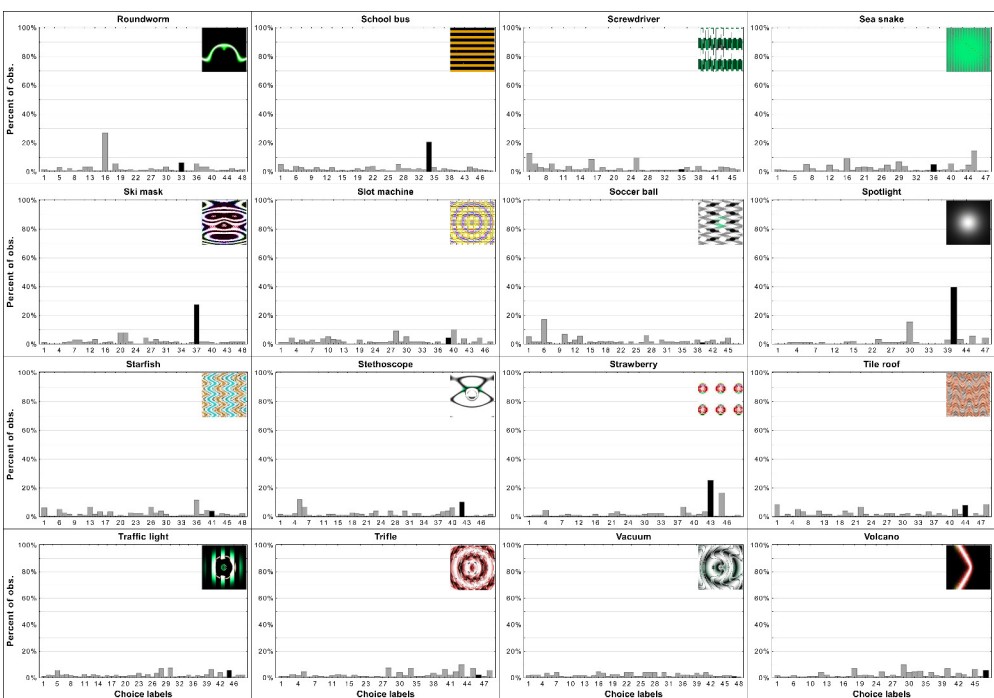

**Appendix 1—figure 6.** Per-item histograms of response choices from Experiment 3b in *Zhou and Firestone, 2019*. Continued.

## Reanalysis of *Zhou and Firestone, 2019*

Roughly similar agreement levels on most images accompanied by mean agreement above chance would indicate some systematic underlying overlap between human and network object recognition. However, as shown in *Appendix 1—figure 1*, there are vast differences in agreement levels depending on adversarial image. The inset of *Appendix 1—figure 1* also shows that the distribution is skewed and that the mean agreement metric overestimates true human-network agreement which is better represented by the median. There is a minority of images which drive agreement as outliers, giving credence to the hypothesis about two separate sources of agreement discussed in the paper.

When trying to determine the nature of human-network agreement it is important to consider distributions of choices on a per-item level not merely whether agreement was above chance for a particular stimulus. If there was general agreement between humans and networks one could expect that for most images the most frequently chosen label would be the one assigned to the stimulus by the network. Additionally, it could be expected that a large percentage of participants choose exactly this label. *Appendix 1—figures 2* and *3* show the top eight most frequently chosen labels by participants in Experiment 3b from *Zhou and Firestone, 2019*. It can clearly be seeen that for only a minority of images the network label was the most often chosen one by participants. Additionally, there is only one stimulus for which majority of participants chose the network label (e.g. 'chainlink fence', image number 6 in the *Appendix 1—figure 2*) while only for a few others can it be said that a fairly substantial percentage of participants chose the network label. For many of the stimuli, the network label is not amongst the top eight choices made by participants. It can also be observed that for most stimuli the top most frequently made choice wasn't overwhelmingly favoured by participants, meaning the distribution of label choices for most stimuli is flat, indicating guessing rather than agreement (*Appendix 1—figures 4–6* for more information).

As was implied in *Appendix 1—figures 2* and *3*, *Appendix 1—figures 4–6* reveal flat distributions of label choices for the vast majority of stimuli. Indeed only eight histograms resemble what could be expected if there was systematic human-network agreement on classification. For those stimuli the most often chosen label was the network label and the percentage of participants who chose the label peaks above the percentage of choices for other labels. There are examples with similar peaks but in which the most often chosen label was not the one assigned to the stimulus by DCNNs. Overall, this provides evidence for the hypothesis that agreement is derived from two

sources. First, some stimuli (e.g. 'chainlink fence') can hardly be called adversarial images since they retain almost all of the features as well as maintain the relationship between features of the target category. In those rare cases, agreement is trivial. In other cases in which agreement is above chance it is likely that those levels were achieved by excluding labels based on a few superficial features. These features were not sufficient for object recognition but did allow for exclusion of labels which do not contain the specific features (e.g. dismissing 'monarch butterfly' when viewing the 'projector' stimulus). We believe that this exact pattern of data supports such a hypothesis.

## Appendix 2

## Supplementary information for Experiments 1–5

| Stimulus | DCNN label | Alternative 1 | Alternative 2 | Alternative 3 |
|---|---|---|---|---|
| | Assault rifle | Accordion | Chainsaw | Piano |
| | Baseball | Bath towel | Parallel bars | Wool |
| | Computer keyboard | Chainlink fence | Crossword puzzle | Honeycomb |
| | Electric guitar | Harmonica | Picket fence | Radiator grille |
| | Freight car | Bubble | Car wheel | Traffic light |
| | King penguin | CD player | Plate | Wall clock |
| | Peacock | Broccoli | Grass snake | Valley |
| | Ski mask | Baboon | Loudspeaker | Totem pole |
| | Tile roof | Rock crab | Shower curtain | Wooden spoon |
| | Volcano | Lighter | Missile | Table lamp |

**Appendix 2—figure 1.** Experiment 1 stimuli and competitive alternative labels.

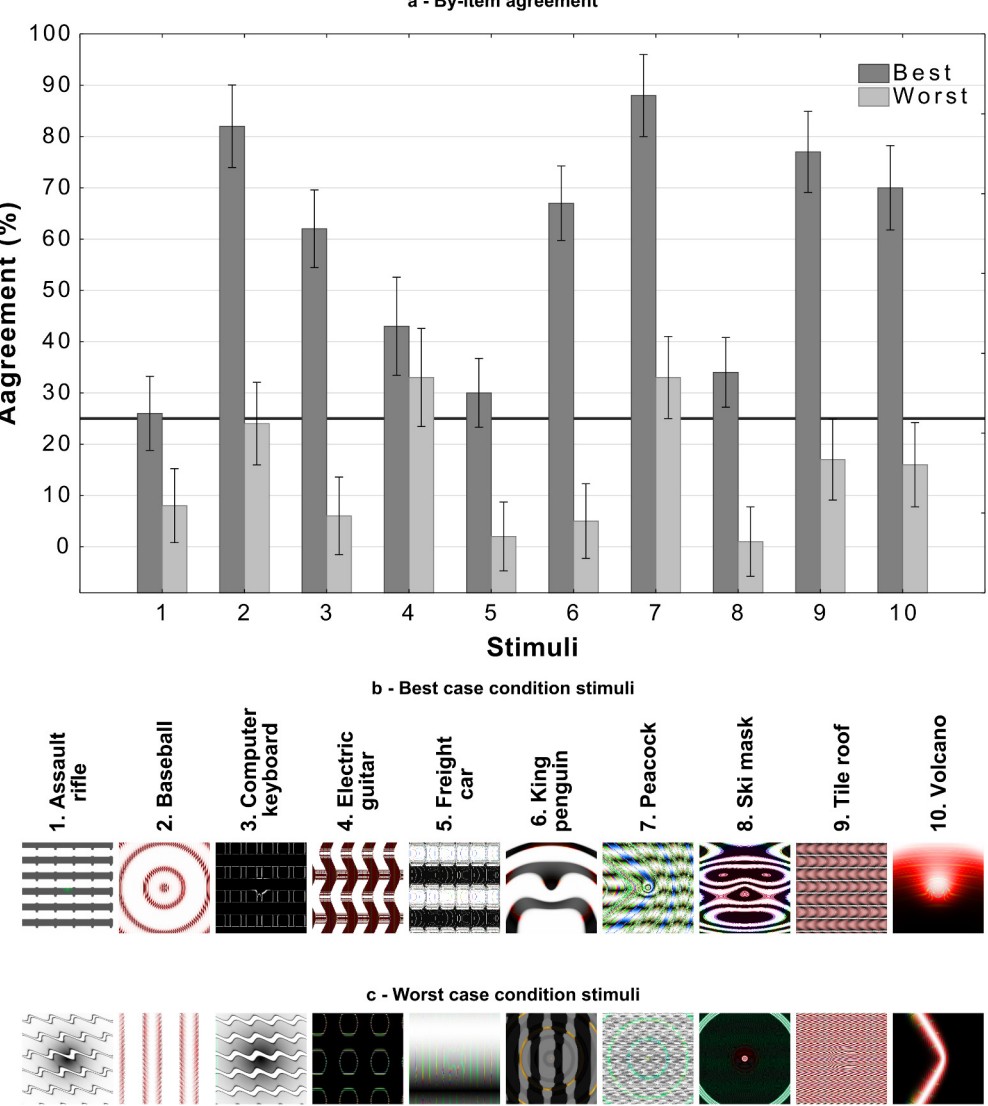

**Appendix 2—figure 2.** An item-wise breakdown of agreement levels in Experiment 2 as a function of experimental condition and category. Average agreement levels for each category in each condition with 95% CI are presented in (**a**) with the black line referring to chance agreement. The best case stimuli are presented in (**b**), these stimuli were judged as containing the most features in common with the target category (out of 5 generated by *Nguyen et al., 2015*). The worst case stimuli are presented in (**c**), these were judged to contain the least number of features in common with the target category.

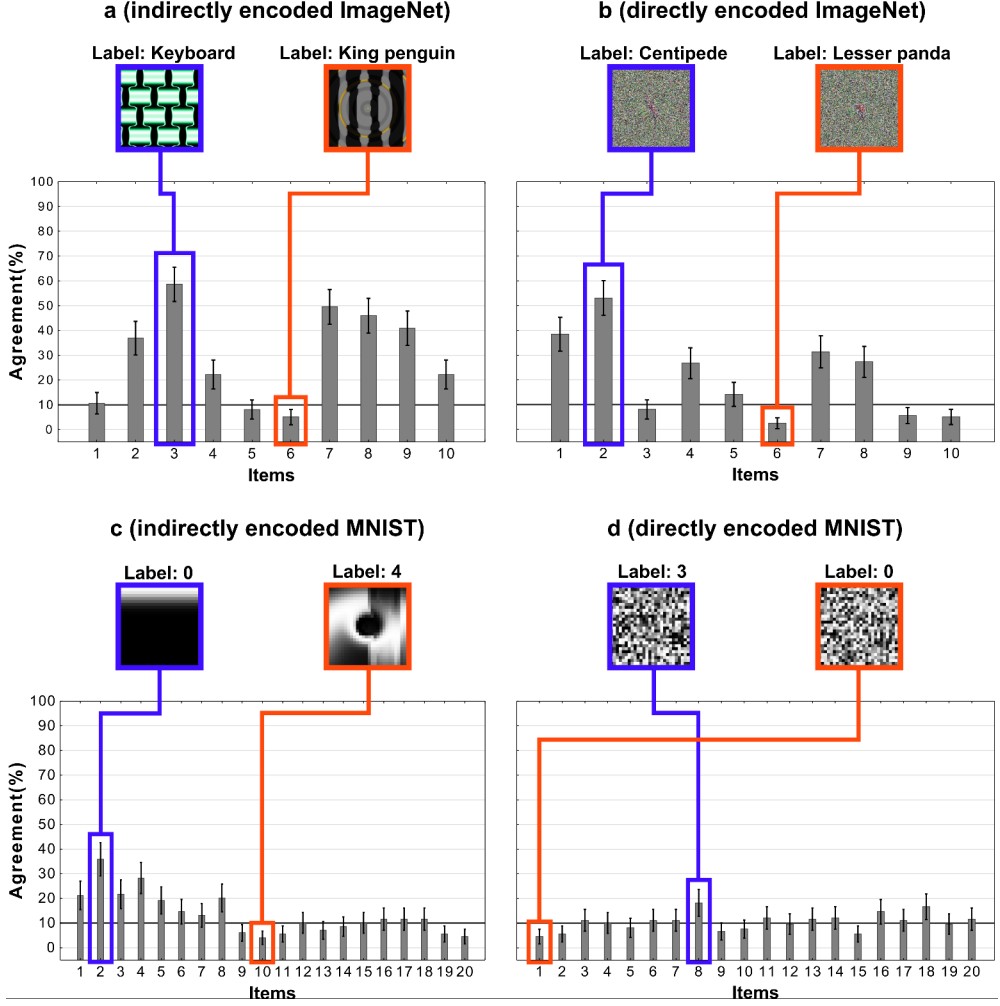

**Appendix 2—figure 3.** An item-wise breakdown of agreement levels for the four conditions in Experiment 3. Each bar shows the agreement level for a particular image, that is, the percentage of participants that agreed with DCNN classification for that image. Each sub-figure also shows the images that correspond to the highest (blue) and lowest (red) levels of agreement under that condition.

| Stimulus | DCNN label | Alternative 1 | Alternative 2 | Alternative 3 |
|---|---|---|---|---|
| | Balloon | Camera | Car wheel | Missile |
| | Chime | Prison | Sliding door | Vault |
| | Digital clock | Mask | Traffic light | Speaker |
| | File cabinet | Computer keyboard | Computer screen | Window shade |
| | Radiator grille | Bottle cap | Manhole cover | Safe |
| | Hook | Cup | Soup bowl | Washbasin |
| | Jellyfish | Bubble | Seashore | Sea snake |
| | Mask | Butterfly | Oil filter | Vase |
| | Matchstick | Bagel | Lighthouse | Orange |
| | Maze | Bubble | King crab | Spider web |

**Appendix 2—figure 4.** Experiment 5 stimuli and competitive alternative labels.

| Stimulus | DCNN label | Alternative 1 | Alternative 2 | Alternative 3 |
|---|---|---|---|---|
| | Menu | Crate | Flatworm | Helix |
| | Mosque | Chainsaw | Computer screen | Missile |
| | Picket fence | Camera | Car wheel | Zebra |
| | Piggy bank | Band aid | Butterfly | Pencil eraser |
| | Soup | Gong | Orange | Potpie |
| | Syringe | Ruler | Safety pin | Street sign |
| | Theatre curtain | Daisy | Microphone | Soccer ball |
| | Vault | Rock crab | Seashore | Volcano |
| | Wardrobe | Camera | Shower curtain | Spotlight |
| | Window screen | Dishrag | Prayer rug | Theatre curtain |

**Appendix 2—figure 5.** Experiment 5 stimuli and competitive alternative labels. Continued.

