## [Decision Letter]

**Acceptance summary:**

The authors address the very timely and important question of whether primate vision functions in a similar manner to modern Deep Learning methods, in particular Deep convolutional neural networks (DCNNs). Several prominent groups have argued that the firing patterns of DCNNs resemble those seen in neural recordings of primate visions, lending some evidence to the idea that DCCNNs and primate vision may be similar. By reanalyzing the data from a previous study claiming that "Humans can decipher adversarial images" and by performing a new set of experiments, the authors show that humans and DCNNs only weakly agree and that agreement is highly dependent on experimental design choices.

**Decision letter after peer review:**

Thank you for submitting your article "What do adversarial images tell us about human vision?" for consideration by *eLife*. Your article has been reviewed by Ronald Calabrese as the Senior Editor, a Reviewing Editor, and three reviewers.

The reviewers have discussed the reviews with one another and the Reviewing Editor has drafted this decision to help you prepare a revised submission.

Summary:

In their paper, Dujmović and collaborators address the very timely and important question of whether primate vision functions in a similar manner to modern Deep Learning methods, in particular Deep convolutional neural networks (DCNNs). There is tremendous interest in this idea. Several prominent groups have argued that the firing patterns of DCNNS resemble those seen in neural recordings of primate visions and lending some evidence to the idea that DCCNNs and primate vision may be similar. One common objection to this analogy is that DCNNs and humans seem to have very different responses to "adversarial mages that fool DCNNs but are uninterpretable to humans." The existence of these adversarial images is often used to argue that DCNNs and primate vision function is very different ways. However, this view was recently challenged by Zhou and Firestone, 2019 that "Humans can decipher adversarial images".

This submission challenges the claim. The manuscript starts by reanalyzing the data from the Zhou and Firestone, 2019 experiments. The authors point out that the original analysis used a summary statistic on individual on responses, rather than the full distribution of responses, arguing that the degree of agreement in the original experiments between DCNN and primate visions was much less than presented in Zhou and Firestone, 2019. In addition, the authors designed four new experiments that used the same basic experimental design as Zhou and Firestone, 2019 but in using a different approach. In particular, in Experiment 1, they address the idea that superficial features were what was driving agreement in the similarity of classifying adversarial examples between DCNNs and humans by using "less adversarial" images. In Experiment 2, they checked whether a subset of categories were represented in a similar manner between humans and DCNNS or if the agreement in Zhou and Firestone, 2019 on certain images was just because a particular adversarial image used by Zhou and Firestone, 2019 was superficially similar to the real image. Finally, they show in Experiments 3 and 4 that it is possible to generate adversarial images that humans and DCNNs recognize in completely different ways.

Essential revisions:

While the reviewers felt that the paper was of sufficient interest for publication (and a fraction were very enthusiastic about its contributions to the literature and its eventual acceptance at *eLife*), there were five key points that would be essential to address before the article could be accepted, enumerated below. Additionally, we also ask the authors to also address the specific comments in the section below.

1) All three reviewers felt strongly that the language/framing of the paper should be tempered to more constructively reflect the relationship of this paper to the literature and to prevent misinterpretation. Specifically, although the discussion about which statistical methods should be computed to support the claims is key to the manuscript and an important point, there are several places (see specific comments for details) where the reader could get the impression that the authors are claiming that Zhou and Firestone, 2019 computed statistical tests incorrectly (rather than taking a different approach than suggested by the authors here) by using language such as ""statistically unsound." Discussion of the question of which tests to compute would be valuable to this field at the interface of ML and psychology, and so we ask the authors to provide clarity to the text accordingly.

2) In their original paper, Zhou and Firestone, 2019 write "We conclude that human intuition is a more reliable guide to machine (mis)classification than has typically been imagined." We ask the authors to comment upon whether that the authors' re-analyses (which still show significantly above-chance classification in Zhou and Firestone, 2019's experiments; e.g., in the authors' Table 2) do support this modest conclusion, even though those analyses are still consistent with a fairly low overall level of human-DCNN agreement.

3) Similarly, the reviewers often had difficulty understanding how subjectively the "best" and "worst" adversarial images were chosen for the experiments? Ideally, this would be done by, say, polling a random group of people, would everyone agree (measured statistically) with these rankings? In absence of such polling, though, we ask the authors to more explicitly discuss which set of adversarial images are used where and to discuss more fully how these choices lead to the conclusions they draw.

4) The authors consider the existence of adversarial images as a strong counterpoint to the possibility that humans and DCNNs categorize images in the same way. An alternative is that, for many cases, humans and DCNNs are categorizing images in similar ways, but that for some edge cases, DCNNs fail, and we have not yet figured out how to build a DCNN that recognizes when it is being fooled. But what does it mean to have the same way of categorizing images? In this paper, the implicit view is that if humans and the DCNN disagree on this suite of particular experiments, then they are working in fundamentally different ways. Under this definition, do all humans categorize images in one particular way? Do all DCNNs, including DCNNs with different architectures? To make their claim, the authors need to set clear criteria by which to judge that image categorization is the same across networks, and show that variability across DCNNs is smaller than the variability between humans and DCNNs.

5) Related, the particular adversarial images to which a network is sensitive are reflective of the training set that determined the DCNN. The DCNN in this study was trained on the ImageNet dataset. How would another artificial network, trained on a different image dataset, classify these adversarial images? This would establish which of their findings are expected of any pair of networks with different training sets.

To address 4) and 5), it would be helpful to observe the performance of additional DCNNs, trained on completely separate image banks, on each of the four experiments. For example, in Experiment 4, participants had to distinguish between adversarial images generated for ResNet-18 and AlexNet. For 3), how well does ResNet-18 identify AlexNet adversarial images? For 4) If you had two instantiations of AlexNet ("AlexNet1" and "AlexNet2") but trained them on disjoint sets of images from the same classes, would AlexNet1 agree with AlexNet2 on adversarial images generated for AlexNet2? If it turns out that, for instance, AlexNet1 agrees with AlexNet2 at chance level, then the interpretation of chance-level human performance is not valid.

Note: comments #12 and onward were written by reviewer #1.

1) DMB's summary of Zhou and Firestone, 2019's result in the Introduction is ambiguous and open to misreading. Readers might hear DMB's paragraph as stating that Zhou and Firestone, 2019 reported that humans agreed with machines 90% of the time they were asked, which of course Zhou and Firestone, 2019 do not report. A more precise and accurate summary of Zhou and Firestone, 2019's result would be something like this:

2) "The authors took a range of published adversarial images that were claimed to be uninterpretable by humans, and in a series of experiments, they showed those images to human subjects next to the DCNN's preferred label and various foil labels. They reported that, over the course of an experimental session, a high percentage of participants (often close to 90%) chose the DCNN's preferred label more often than would be expected by chance."

3). The final sentence of the Abstract needs to be tempered somewhat. DCNNs are a "promising model," in particular for many aspects of lower-level vision, but they are not a complete model.

4) Introduction: Correct the spelling of inferotemporal cortex.

5) Introduction: The paper revolves on the analysis of the classification of adversarial images, yet beyond showing examples of types of adversarial image, there is not a clear definition of what an adversarial image is and to what extent adversarial images are classified into types (e.g., 'fooling' or 'naturalistic'). Are these all the types of adversarial images?

6) Subsection “Reassessing the level of agreement in Zhou and Firestone (2019)” and Materials and methods section: Why are choices are independent in the null model? Was the task constructed so that choices may repeat? If so, then this is fine. On the other hand, if only 48 images and 48 labels are presented, a participant may treat this as a 'matching' task inducing correlations between choices. For 48 items, the correction to the null distribution of number of correct choices is minor, but it would not be for a smaller number of items. Either way, a clearer description of the task design is warranted.

7) Subsection “Reassessing the level of agreement in Zhou and Firestone (2019)”and Table 1: Please explain what the 'Images' column means in this table. Up to this point, the text has referred only to the participant-level analysis.

8) Subsection “Reassessing the basis of the agreement in Zhou and Firestone (2019)”: grammar – missing definite article

9) Subsection “Reassessing the basis of the agreement in Zhou and Firestone (2019)”: grammar – colloquial usage

10) Discussion section: This final conclusion, that DCNNs that also incorporate a number of additional features would be less sensitive to adversarial attacks, is interesting. I would recommend reading and possibly citing work on "capsule networks" from Hinton/DeepMind.

11) Figure Appendix 1—figure 4: check cross-reference – there's a "?"

12) Critiques of scientific research, including this one, are important, valuable, and welcome. At the same time, a core obligation of any such critique is to present its target charitably, and to engage its strongest arguments. I'm worried that this paper does not do that in its current form. A key piece of context that, in my opinion, DMB's paper loses sight of is that Zhou and Firestone, 2019 were directly motivated by Nguyen et al.'s statement that their fooling images were "completely" and "totally unrecognizable to human eyes" – emphasis on "completely" and "totally". Zhou and Firestone, 2019 noticed that many of the images did seem to have some features in common with their target class, and that Nguyen et al. had collected no human data on this; and so Zhou and Firestone, 2019 simply set out to show that the fooling images in Nguyen et al., were not, as a rule, "totally unrecognizable". It's my strong opinion, even after reading and accepting much of what DMB say, that Zhou and Firestone, 2019's data and analyses (and even DMB's reanalyses) still support Zhou and Firestone, 2019's modest conclusion, and I strongly encourage the authors to consider the same. Note that even Nguyen himself agreed; he wrote to us after reading Zhou and Firestone, 2019's paper to say: "After reading this paper, I develop a stronger intuition that humans can decipher very well those robust, and model-transferrable AXs" and that "I personally agree that the phrase 'totally unrecognizable to human eyes' might have been an overstatement. Thanks to your work for pointing it out!". I agree with that statement too.

Indeed, throughout the paper, DMB write statements like Zhou and Firestone, 2019's "conclusions are not justified" [Introduction], but without actually stating what Zhou and Firestone, 2019's conclusions are. In fact, Zhou and Firestone, 2019's conclusions are clear and explicit:

- "We conclude that human intuition is a more reliable guide to machine (mis)classification than has typically been imagined" (final paragraph of Zhou and Firestone, 2019's Introduction, where "than has typically been imagined" refers to Nguyen et al.'s claim).

- "This implies at least some meaningful degree of similarity in the image features that humans and machines prioritize – or can prioritize – when associating an image with a label" (first paragraph of Zhou and Firestone, 2019's Discussion).

and

- "The present results suggest that this particular challenge to notions of human-machine similarity may not be as simple as it appears (though there may of course be other reasons to doubt the similarity of humans and machines)" (first paragraph of Zhou and Firestone, 2019's Discussion section).

In other words, Zhou and Firestone, 2019's conclusion is not that there's a high level of agreement (though see more on that below), or that DCNNs are good models of human vision, etc.; it is simply that the images are not "totally unrecognizable" in the way Nguyen et al. had claimed. And there can be no doubt that this is Zhou and Firestone, 2019's conclusion, since the only time Zhou and Firestone, 2019 even use the word "conclude" is in the above quote.

It is my strong opinion, perhaps my strongest opinion throughout this review, that DMB have not painted that conclusion in an accurate, fair, or charitable way. And so it is also my very strong opinion that the paper must be revised to state that conclusion accurately, and to consider whether the data as a whole (including Zhou and Firestone, 2019's and DMB's analyses) truly are inconsistent with it. Human-machine agreement can be low (as DMB suggest) while still being consistent with Zhou and Firestone, 2019's conclusion that the images are not "totally unrecognizable". There is not as much disagreement between Zhou and Firestone, 2019 and DMB as DMB makes it seem, and there's no reason not to acknowledge that agreement when it exists.

13) The authors claim that Zhou and Firestone, 2019 reported a "surprisingly large agreement between humans and DCNNs" [Introduction]; "the reported level of agreement is nearly 90%" [Results section]. These statements are inaccurate, or at least imprecise. Zhou and Firestone, 2019 do not report a "level of agreement of nearly 90%": "level of agreement" is a new phrase that DMB use (appearing nowhere in Zhou and Firestone, 2019's paper) that makes it seem as though Zhou and Firestone, 2019 reported something they did not, and that also makes it seem that Zhou and Firestone, 2019's measure is aimed at the same quantity as DMB's measure. Additional clarity here would be appreciated, since this comes up later (e.g., in saying that the agreement was "lower than reported", which I worry is misleading – the two measures simply report different quantities).

14) Similarly, DMB wait too long to note that Zhou and Firestone, 2019 already computed DMB's "average agreement" measure; indeed, it was the very first analysis in Zhou and Firestone, 2019's paper, but DMB only later acknowledge this, instead starting with Zhou and Firestone, 2019's Experiment 3 two pages earlier. In Experiment 1, Zhou and Firestone, 2019 state: "Classification "accuracy" (i.e., agreement with the machine's classification) was 74%, well above chance accuracy of 50% (95% confidence interval: [72.9, 75.5%]; two-sided binomial probability test: p < 0.001)". Indeed, this alone shows how it is misleading of DMB to claim that Zhou and Firestone, 2019 report a "level of agreement" near ceiling; Zhou and Firestone, 2019 are clear that "agreement with the machine's classification" is 74%, just as DMB report.

15) I stated above that I still think Zhou and Firestone, 2019's analyses support their conclusions. To see why, consider an analogy. Suppose someone claimed that dieting is "completely" and "totally" ineffective for losing weight, as Nguyen et al., claimed that their fooling images were "totally unrecognizable to human eyes". If you were skeptical (as Zhou and Firestone, 2019 were), you could assign subjects to a diet and see if they lose weight. There are then two standard approaches to evaluating whether the diet was "totally ineffective". You could (1) ask *how much weight* the cohort lost as a whole, and whether that amount deviates significantly from what would be expected under the null; or you could (2) ask *how many people* lost weight, and ask whether that number significantly deviates from what would be expected under the null. For example, you could (1) discover that the cohort lost 10% of their bodyweight on average, and that this number deviated significantly from 0%; or you could (2) discover that 95% of dieters lost some amount of weight (whether that amount was a lot or a little), and that this number significantly deviated from 50%.

Zhou and Firestone, 2019 take both approaches – (1) and (2) – in their Experiment 1, but then settle on (2) for the rest, because it seemed better suited to refute the claim that adversarial images are "completely unrecognizable to humans". By contrast, (1) is an approach DMB prefer, as is evident from Table 2. That's fine; as DMB show in that table, it also confirms Zhou and Firestone, 2019's conclusions by showing significantly above-chance average agreement. (I comment on Table 1 below.) What should be clear, however, is that these two approaches are both valid, and indeed have certain strengths and weaknesses over one another. For example, if approach (1) finds that subjects lost 10% of their bodyweight on average, that still might not refute the claim that the diet is ineffective, since it could be that most subjects gained a small amount of weight while a minority of subjects lost a very large amount of weight – a possibility that would make this measure ill-tailored to refuting claims of ineffectiveness (since it causes weight gain in most people). Conversely, if you used approach (2) and found that 95% of people lost some amount of weight, then that wouldn't tell you how much weight the cohort (or any one person) lost; but it is still informative, because if 95% of people lose weight on a diet (whether that amount is 1 pound or 50), then whatever you think of this diet, it's hard to plausibly claim that it's "completely" and "totally" ineffective (even if it might not be very effective overall). And, of course, both approaches combined would be best. But, again, they're both valid: If 98% of subjects pick the machine's label numerically more often than not, and if 98% is significantly higher than 50%, then it just can't be true that these images are "completely" and "totally" undecipherable – even if any given subject is only agreeing a bit, and even if that agreement is not even statistically significant within a single subject (just like one needn't say of any given dieter that they lost a "significant" amount of weight; if 95% of dieters lose weight, then that is telling, as long as 95% significantly differs from 50% in the sample).

So DMB are being too critical. There are multiple ways to understand these data, depending on the researchers' goals. DMB are correct to say that Zhou and Firestone, 2019's analyses in Experiment 3 "assign the same importance to a participant that agrees on 2 out of 48 trials as a participant who agrees on all 48 trials with the DCNN" [84]; but this just isn't a problem for Zhou and Firestone, 2019's research question. Zhou and Firestone, 2019's goal was to refute the claim that the fooling images in Nguyen et al. were "completely unrecognizable to humans". Perhaps DMB do not share this goal; but that is no reason to call Zhou and Firestone, 2019's approach "misleading and statistically unsound". Both Zhou and Firestone, 2019's approach and DMB's approach are fine. And, again, *both* approaches ended up coming out Zhou and Firestone, 2019's way, as DMB confirm in Table 2.

The paper should be revised to reflect this. There is not actually a large disagreement here between Zhou and Firestone, 2019 and DMB as regards to how Zhou and Firestone, 2019's data interact with Nguyen et al.'s claim (which is what Zhou and Firestone, 2019 were after). To give an example (though of course DMB could use language other than mine), the paper could say something like "Whereas the analyses Zhou and Firestone, 2019 use may have been sufficient to refute previous claims that such images were "totally unrecognizable to human eyes", those analyses are still consistent with a fairly low overall level of human-machine agreement." (But of course, much else would then have to change too, since DMB are so persistent in calling Zhou and Firestone, 2019's conclusions unjustified.)

16) A central issue with the re-analyses is how much DMB's discussion focuses on Zhou and Firestone, 2019's Experiment 3b. In my opinion, this was a strange experiment to choose, especially when it comes to giving a charitable critique. The entire purpose of Zhou and Firestone, 2019's Experiment 3b was to make human-machine agreement as difficult as possible, and this is the single least-discussed experiment (of 8) in Zhou and Firestone, 2019's entire paper – not the cornerstone of Zhou and Firestone, 2019's claims in any way. When Zhou and Firestone, 2019 observed above-chance human-machine agreement in Experiment 1, there was a worry that above-chance agreement when given two alternatives wasn't so impressive. So, Experiments 3a and 3b presented all the labels for every image at once, to see if subjects could show above-chance agreement even in extremely taxing circumstances. Zhou and Firestone, 2019 were very explicit about this: Zhou and Firestone, 2019 say "These results suggest that humans show general agreement with the machine even in the taxing and unnatural circumstance of choosing their classification from dozens of labels displayed simultaneously". So Zhou and Firestone, 2019 already described this task as "taxing and unnatural" and unlikely to produce high human-machine agreement. And indeed it's certain that subjects do not actually read all the labels before making judgments, because they often answer in just a few seconds – not enough time to have looked at all the labels. So, of course agreement will be low – that was the point of that experiment.

Indeed it is possible to see here – http://www.czf.perceptionresearch.org/adversarial/expts/texture48.html – how imposing that is, whereas Experiment 1 – here http://www.czf.perceptionresearch.org/adversarial/expts/texture.html – feels much more natural. Yet, subjects in Experiment 3a and 3b still showed a mean agreement of ~10%, rather than the chance level of ~2%. We thought that was impressive, given the circumstances. DMB may disagree, but that is simply a difference in opinion or taste, not an undermining of Zhou and Firestone, 2019's claims. (Indeed, DMB later describe this result as "striking"; strikingly high? If so, why not say so earlier?)

To be fairer and more accurate, the relevant claims in DMB's paper should also include Zhou and Firestone, 2019's Experiment 1 as an example. For example, subsection “Reassessing the level of agreement in Zhou and Firestone (2019)” should first describe how Zhou and Firestone, 2019 analyzed Experiment 1, and then describe Experiment 3. Figure 1.1 should include Experiment 1 and Experiment 3. And so on. Otherwise, I worry that DMB might be cherry picking, choosing the examples from Zhou and Firestone, 2019 that they feel are weakest, rather than the ones they feel are strongest. For example, DMB state in subsection “Reassessing the basis of the agreement in Zhou and Firestone (2019)” that "For the rest of the images agreement is at or below chance levels". But that's only true of Experiment 3; for Zhou and Firestone, 2019's Experiment 1, which was not designed to be so taxing on subjects, even DMB's binomial analysis shows that agreement is significantly above chance on over 85% of images. It is potentially misleading (if not simply false) to claim that "For the rest of the images agreement is at or below chance levels". This discussion should be more balanced, and either focus on Experiment 1 or at least include parallel analyses for Experiment 1's results whenever DMB discuss Experiment 3.

17) Beyond the re-analyses, I'm worried that DMB present their views as alternatives to Zhou and Firestone, 2019, when in fact many of these views were already explicitly articulated by Zhou and Firestone, 2019. I've already emphasized how this is true of Zhou and Firestone, 2019's main conclusion. But another example is DMB's suggestion that human-machine agreement on these images reflects "participants making educated guesses based on some superficial features (such as colour) within images and the limited response alternatives presented to them" [Introduction]. Truly, this is already Zhou and Firestone, 2019's exact view about those experiments. Again, with quotes, Zhou and Firestone, 2019 say that subjects "may have achieved this reliable classification not by discerning any meaningful resemblance between the images and their CNN-generated labels, but instead by identifying very superficial commonalities between them (e.g., preferring "bagel" to "pinwheel" for an orange yellow blob simply because bagels are also orange-yellow in color)". That applies to Experiments 1 and 3, which use the same labels.

DMB subtly acknowledge this elsewhere, but otherwise they present this hypothesis as if it is original to them. Consider subsection “Reassessing the basis of the agreement in Zhou and Firestone (2019)” or subsection “Experiment 5: Transferable adversarial images”, which says of Elsayed et al., that "These findings are consistent with *our observation* that some adversarial images capture some superficial features that can be used by participants to make classification decisions" (emphasis added). But this is already Zhou and Firestone, 2019's exact hypothesis. Both Zhou and Firestone, 2019 and DMB believe subjects are making educated guesses. (Indeed, Zhou and Firestone, 2019 think that may be the right way to describe what the CNN is doing too: Zhou and Firestone, 2019 write "both the CNNs' behavior and the humans' behavior might be readily interpreted as simply playing along with picking whichever label is most appropriate for an image", and "CNNs are.… forced to play the game of picking whichever label in their repertoire best matches an image (as were the humans in our experiments)".

Perhaps "consistent with Zhou and Firestone, 2019's and our observation" would be a better statement for this and other examples. But it's simply not right to portray this as DMB's own or original interpretation; it is precisely Zhou and Firestone, 2019's interpretation, as these quotes show. This is another example where DMB portray disagreement that may not really exist, and so should attribute this interpretation to Zhou and Firestone, 2019.

18) "Using a statistic that treats 45% agreement as chance is liable to be misinterpreted by readers" [subsection “Reassessing the level of agreement *in Zhou and Firestone (2019)”*]. Perhaps this is true; I agree that Zhou and Firestone, 2019 could have been clearer. But doesn't this support the *opposite* of DMB's point? The reader that DMB are imagining would normally take chance to be 50% (as Zhou and Firestone, 2019's figures show, perhaps unclearly), whereas the true value of above-chance-performing participants is 88% or 90%. So the fact that chance is 45% rather than 50% makes the result *more* impressive, not less impressive, since 88% is even farther above chance than the reader was led to believe.

19) In the very next paragraph, DMB make the exact kind of misleading statement that they just charged Zhou and Firestone, 2019 with, and in a way that, I worry, makes this part of the reanalysis potentially misleading. DMB say "Measured in this manner (independent 2-tailed binomial tests with a critical p-value of 0.05 for each participant), the agreement in Experiment 3a between DCNN and participants drops from 88% to 57.76%" [Results section]. But in fact there is no "drop", because *chance* changes across these two measures. For the 88% measure, chance is 45%; but for the 57.76% measure, chance is 5% (or even 2.5%), since that's the α value of the test that DMB run. In other words, you would expect 5% of subjects to perform significantly different from chance under the null. So DMB have moved the standard without properly alerting the reader. Indeed, 57.76% when chance is 5% is, if anything, *more impressive* than 88% when chance is 50%. This paragraph should be revised to reflect this. It could say something like, "Measured in this manner, the agreement in Experiment 3a between DCNN and participants moves from 88% (where chance is 45%) to 57.76% (where chance is 5%)"; it would then be clear that "drops" isn't appropriate. They should also report this for Experiment 1, in line with my comment #3 above.

20) This same issue appears in Table 1. To be more interpretable, that table should not only report these %s but should also report the chance level for each %, just as DMB do for Table 2. That will help the reader understand how to interpret what DMB call the "drop" from higher numbers to lower numbers. I strongly recommend this; the table is, I worry, highly misleading without this correction – i.e., a column for something like "what would be expected by chance" (which, I gather is 5%). Indeed, a problem with the analysis in Table 1 is that, in some sense, there's no "null hypothesis"; there's no standard by which the analysis can decide whether the cohort of images was "totally unrecognizable" or not. The value of Zhou and Firestone, 2019 analysis, and DMB's Table 2, is that it's clear how to reject the null hypothesis: i.e., if significantly more images are numerically above chance than are numerical below chance (Zhou and Firestone, 2019), or if mean agreement is significant above chance (Zhou and Firestone, 2019 and DMB). But DMB's Table 1 analysis doesn't really work that way. So that's why it's crucial to portray what chance is for those tests, in the table itself, so that readers can see that it was 85% agreement when chance was 5%, or 57% when chance was 5%, etc.

21) A final issue with the reanalyses themselves, and in some ways the biggest one, is just that Zhou and Firestone, 2019's conclusions survive them, in ways that make it extremely confusing to me why DMB draw the conclusions they do. As Table 2 shows, every one of Zhou and Firestone, 2019's experiments continues to show significantly above-chance human-machine agreement, even on the analytical approach that DMB prefer. And DMB acknowledge this: They say "it is nevertheless the case that even these methods show that the overall agreement was above chance" [subsection “Reassessing the basis of the agreement in *Zhou and Firestone* (*2019*)”]. First, DMB should not wait this long to say so; they should say as early as possible that their analyses, like ours, show significantly above-chance agreement. But second, this demonstrates that Zhou and Firestone, 2019's conclusions, as stated above, are secure after all. I must repeat again that Zhou and Firestone, 2019's claims, made explicitly in their paper, are as follows: "that human intuition is a more reliable guide to machine (mis)classification than has typically been imagined", and that the results "impl[y] at least some meaningful degree of similarity in the image features that humans and machines prioritize-or can prioritize-when associating an image with a label". DMB confirm that these conclusions are even more robust than Zhou and Firestone, 2019 suggested, since they are supported by Zhou and Firestone, 2019's approach and DMB's.

22) Moving to the experiments: DMB's abstract states that "it is easy to generate images with no agreement" [Abstract]. I don't see how this claim is justified by their experiments. First, half or more of DMB's experiments reflect a failure to "generate images with no agreement". Experiment 1 shows above-chance agreement: "A single sample t-test comparing the mean agreement level to the fixed value of 25% did show the difference was significant (t(99) = 3.00, p = .0034, d = 0.30)" [179]. Experiment 2 does so as well: when the best and worst cases are combined, their average agreement is above chance – it's only through picking images that DMB think look undecipherable that they were able to find undecipherable images (note that Zhou and Firestone, 2019 never state that all images will be undecipherable, but rather that the procedure for producing such images will tend to produce decipherable images in general; Experiment 2 confirms this). Experiment 3 does so as well: "Participants were slightly above chance for indirectly-encoded MNIST images (t(197) > 6.30, p <.0001, d = 0.44)" [subsection “Experiment 3: Different types of adversarial images”]. (Only Experiment 4 does not; I'll return to that later.)

This was confusing. Why, if it is "easy" to generate images with no agreement, did DMB so frequently fail to do so? This conclusion should be altered to something like "While difficult, it is possible to generate images with no agreement"; but surely not "easy"!

23) Experiment 1 is hard to interpret as run; or, if it is, it doesn't show a lack of human-machine agreement. The authors chose labels that they subjectively felt were good competitors for the DCNN's label, and found that agreement dropped but was still significantly above chance. This is unsurprising and no threat to Zhou and Firestone, 2019's view, for at least three reasons.

First, this conclusion was already explicitly reached by Zhou and Firestone, 2019, who also considered this and whose Experiment 2 showed that human-machine agreement drops with more competitive labels. DMB mention Zhou and Firestone, 2019's Experiment 2, and criticize it on other grounds; but those grounds are independent of this aspect of Zhou and Firestone, 2019's discussion. In other words, even granting DMB's criticism of Zhou and Firestone, 2019's Experiment 2, Zhou and Firestone, 2019 still perfectly anticipate the conclusion of DMB's Experiment 1. But again, as above, DMB do not credit Zhou and Firestone, 2019 with this, and instead present this as though it is original to DMB. DMB should acknowledge that their Experiment 1 confirms the results of Zhou and Firestone, 2019's Experiment 2 – that more competitive labels should reduce agreement.

Second, DMB's version (but not Zhou and Firestone, 2019's version) is much more difficult to interpret because it is almost a form of "double-dipping": The researchers, DMB, are human beings with visual systems who can appreciate which images look least like certain labels; so they picked the foil labels they thought fit best, and then discovered that other humans (their subjects) agreed. But Zhou and Firestone, 2019 never claimed that humans would pick the DCNN's label as the literal #1 label among all 1000! Again, Zhou and Firestone, 2019's claim is only that humans favor the DCNN's label better than would be expected if the images were "totally unrecognizable to human eyes". So, this experiment is perfectly consistent with Zhou and Firestone, 2019's view. I keep returning to this because DMB have made Zhou and Firestone, 2019's paper the cornerstone of their own paper. If, instead, they just presented four new interesting experiments, they could interpret those experiments in their chosen way, and readers could decide to be persuaded or not. But instead, DMB present these experiments as refuting Zhou and Firestone, 2019. In that case, they have to get Zhou and Firestone, 2019's claims right. All Zhou and Firestone, 2019 need is for the humans to think the DCNN's label fits better than previously thought; Zhou and Firestone, 2019 don't need it to be the very best label.

Third, and most crucially, DMB's Experiment 1 still shows above-chance agreement! So it seemingly shows the opposite of DMB's claim (that their experiments show it is easy to generate images with "no agreement"), and continues to support Zhou and Firestone, 2019.

24) Experiment 2 is also hard to interpret, for the same reason as Experiment 1, and for an additional reason. It shows that some images are easy to decipher (showing above chance classification), but some are hard to decipher (showing below chance classification). Note that Zhou and Firestone, 2019 explicitly predict this, but again DMB fail to acknowledge this. Zhou and Firestone, 2019 state: "A small minority of the images in the present experiments.… had CNN-generated labels that were actively rejected by human subjects, who failed to pick the CNN's chosen label even compared to a random label drawn from the image set. Such images better meet the ideal of an adversarial example, since the human subject actively rejects the CNN's label". So it is no surprise that it is possible to find images of the "worst-case" type by having a human pick them out; Zhou and Firestone, 2019 already said that should happen, when they wrote "An important question for future work will be whether adversarial attacks can ever be refined to produce only those images that humans cannot decipher, or whether such attacks will always output a mix of human-classifiable and human-unclassifiable images; it may well be that human validation will always be required to produce such truly adversarial images". DMB's Experiment 2 is perfect evidence for Zhou and Firestone, 2019's prediction; they show, just as Zhou and Firestone, 2019 say, that a process that tends to generate decipherable images will also generate some undecipherable ones that a researcher could pick out. This is yet another example of DMB making a claim as if it is original to them, rather than crediting Zhou and Firestone, 2019 with that claim and noting that DMB's results *confirm* Zhou and Firestone, 2019's claims or predictions. It continues to confuse me why DMB frame things this way. There is no need to write as though there is a disagreement here.

25) Indeed, a relevant difference between Zhou and Firestone, 2019's experiments and DMB's Experiment 2 is that Zhou and Firestone, 2019 used images that they didn't even "choose"; they simply used the ones Nguyen et al. displayed in their paper. DMB show that, if the researcher selects a subset of those images with the intent to pick the undecipherable ones, then it is possible to do so. But that's not what's at issue; what's at issue is whether the algorithmic process that generates such images tends to produce only undecipherable images, or a mix of decipherable and undecipherable images. Zhou and Firestone, 2019 chose their images in an unbiased/random way (at least, relying on Nguyen et al.'s presentation), and found decipherability. That's a crucial difference: "unbiased" vs "biased" selection of images.

26) Can the authors make their experiment code available? I apologize if I missed it, but I only saw the data and images in their OSF archive.

27) Experiment 3 also contracts DMB's stated conclusions, and confirms Zhou and Firestone, 2019's hypothesis yet again; even though "To our [DMB's] eyes, these MNIST adversarial images looked completely uninterpretable" [subsection “Experiment 3: Different types of adversarial images”], they still *were* interpretable! As DMB state, "Participants were slightly above chance for indirectly-encoded MNIST images (t(197) > 6.30, p <.0001, d = 0.44)" [subsection “Experiment 3: Different types of adversarial images”]. Indeed, DMB later contradict this result by saying "Experiment 3 showed that it is straightforward to obtain overall chance level performance on the MNIST images"; but that is not what happened, as subsection “Experiment 3: Different types of adversarial images” shows. It was not straightforward; the images, as a group, were deciphered above chance.

28) Another issue with Experiment 3 is that the study used some labels that would be highly unfamiliar to subjects (or, at least, it seems to have done so; again, the experiment code would be helpful). For example, Appendix 2—figure 3 highlights that subjects were reluctant to call a certain image a "Lesser Panda". The authors seem to interpret this as meaning that subjects believed the image did not look like a Lesser Panda. But, of course, an alternative is that the subjects don't know what a Lesser Panda is. Isn't this an alternative explanation? If so, it would have nothing to do with decipherability, and instead to do with whether subjects know what certain ImageNet labels refer to. Indeed, ImageNet gives "Red Panda" as an alternative, but DMB chose to use "Lesser Panda"; why? The image contains a central red patch; I'd strongly predict that subjects would have classified it above chance if DMB hadn't chosen the much more obscure label "Lesser Panda".

29) Experiment 4 is the most interesting contribution of the paper. Indeed, considering everything I have written above – which I acknowledge has been quite negative – Experiment 4 seems interpretable and really does show chance-level classification. This is the part of the paper that could make a new and meaningful contribution, in way that is not misleading, does not misconstrue Zhou and Firestone, 2019's conclusions, and does not show above-chance deciphering. Zhou and Firestone, 2019 do have a reply to this – it is a version of the "acuity" reply, which DMB consider for their indirectly encoded images (and rightly reject), but not for their directly encoded ones. DMB cite evidence that suggests this: Elsayed et al., showed that when properties of the primate retina are incorporated into a CNN that is adversarially attacked, those attacks do look quite decipherable to humans. But the fact that we would give this reply doesn't really bear on this review of DMB's paper, or its publishability. So even though I disagree with DMB's interpretation of Experiment 4, I have no problem with it in the way I do with the rest of the paper.

30) DMB's item-level analysis was described in a way I found confusing or maybe even misleading. Consider subsection “Reassessing the basis of the agreement in *Zhou and Firestone* (*2019*)”: "agreement on many images (21∕48) was at or below chance levels. This indicates that the agreement is largely driven by a subset of adversarial images". But what DMB call a "subset" was in fact a majority of images! 27/48 here, and 41/48 in Experiment 1. (Indeed, it's not made explicit where this number comes from; in Experiment 3, 39/48 are numerically above chance and 9/48 are numerically below chance. The authors should clarify when they are referring to numerically above chance and when significantly above chance, and to always report what chance is for these analyses.) Again, I don't mean to say this disrespectfully, but it frequently feels that DMB are going out of their way to describe Zhou and Firestone, 2019's results in uncharitable ways. DMB note that 85% of images in Experiment 1 are significantly agreed-with above chance, and that 57% are significantly above chance in Experiment 3b. (And if they combined the data from 3a and 3b, which they do elsewhere, they would find that this total is closer to 70%). Calling a majority of images (85%, 70%, or 57%) a "subset" is of course literally true, but it implies that it's somehow a small number of images, when it fact it's most of the images! Especially when chance for all of these statistics is only 5%. Please refer to it that way, rather than imply it's some kind of small number.

30) Please also annotate the lines in Appendix 1—figure 1, with labels, so that it is immediately clear to the reader that the black line is chance.

31) The authors should always make clear when the stimuli they show to readers were chosen algorithmically or chosen by the authors' own subjective impression of them. For example, Figure 3 could give the impression that authors have some procedure to generate best-case and worst-case images; indeed, I originally interpreted the figure that way. But in fact, I now understand that this just reflects their own choices about which images look least like their target class. So this caption should say so – something like, "Example of best-case and worst-case images for the same category ('penguin'), as judged by the present authors, and as used in Experiment 2". And so on elsewhere, including the generation of labels. Another example is Appendix 2—figure 2, which says "these were judged to contain the least number of features in common with the target category". It would be clearer to say "we judged these images to contain the least…". I know this does happen in some places, but even there it is confusing (for example, DMB say they picked images from "each category"; but in fact I believe it's just each image from one of 10 categories, right? It's worth being especially clear here on both counts).

32) Similarly, in subsection “Experiment 1: Response alternatives” ("We chose a subset of ten images from the 48 that were used by Zhou and Firestone, 2019 and identified four competitive response alternatives (from amongst the 1000 categories in ImageNet) for each of these images"), the authors should state the procedure they used to do this. Why did they choose only 10 images? How did they choose the response alternatives? They give some examples of their negative criteria (e.g., no semantic overlap), but that still leaves out a lot of the selection process. Of course, if the answer is just that they selected in advance the images and labels that they thought would show low agreement, it's important to say so explicitly (though that would, of course, undermine aspects of the experiment's interpretation). I'm also confused why DMB excluded "basketball" from the "baseball" image; they are both kinds of balls, but of course people would be unlikely to visually confuse them – isn't this a (likely unintentionally) self-serving experimental decision?

33) I may well have an overly sensitive ear here, but there is a feeling throughout the paper that DMB believe Zhou and Firestone, 2019 behaved in a sneaky or obfuscatory way in reporting their results. Multiple colleagues have shared with me a similar reading of DMB's paper (after seeing their publicly posted preprint), wondering why it is so sharply worded and insinuatory. I have to say I agree. This is especially unfortunate because nothing could be farther from the truth: Zhou and Firestone, 2019 were transparent about all of these analyses, and just in case we weren't, we proactively made all of our data publicly available so that researchers could know exactly what we did – that, of course, is how DMB acquired the data in the first place. (DMB do not mention this either; it would be informative to the reader, and perhaps more collegial of DMB, to state that the reason they were able to reanalyze Zhou and Firestone, 2019's data was because of Zhou and Firestone, 2019's proactive transparency in making them public.) I hope that a revision, whether here or elsewhere, can be more respectful of other researchers' motivations and not insinuate hidden analyses or selective reporting.

For me, the tone is present throughout, such that it is hard to point out every example. Here are some:

- "Our first step, in trying to understand the surprisingly large agreement between humans and DCNNs observed by Zhou and Firestone, 2019, was to reassess how they measured this agreement" [subsection “Reassessing the level of agreement in *Zhou and Firestone* (*2019*)”]. But there was no need for DMB to find themselves "trying to understand" these analyses, as if those analyses were somehow obscure or hidden; the analyses, code, and data were made publicly available alongside the paper itself.

- "We noticed that Z and F used images designed to fool DCNNs trained on images from ImageNet, but did not consider the adversarial images designed to fool a network trained on MNIST dataset" [subsection “Experiment 3: Different types of adversarial images”]. Again, DMB write as if they are suspicious or something. But the reasons are simply that (a) Nguyen et al., highlight their ImageNet images much more (e.g., in their Figure 1), and (b) MNIST-trained networks aren't usually claimed to resemble human vision in the same way as ImageNet-trained networks. Moreover, Zhou and Firestone, 2019 *do* "consider the adversarial images designed to fool a network trained on MNIST dataset"; that's Zhou and Firestone, 2019's Experiment 5. So this language is not only unnecessary, but also even false.

- "when we examined their results more carefully, the level of agreement was much lower than reported" [Discussion section]. There are two problems here. First, what does "more carefully" mean in this context? More carefully than Zhou and Firestone, 2019? That really seems to imply that Zhou and Firestone, 2019 made an error or something, which DMB do not in fact believe as far as I know. DMB simply prefer another measure, not a "more careful" one, and as DMB acknowledge, Zhou and Firestone, 2019 already carry out some of their preferred analyses. And "much lower than reported" is simply inaccurate; it's fine that DMB prefer a different measure, but that's not the same as Zhou and Firestone, 2019 falsely or inaccurately reporting theirs. All instances of "lower than reported" simply must be revised; Zhou and Firestone, 2019 reported everything accurately – DMB just prefer a different analysis.

- The editor and other reviewers have also flagged "statistically unsound" and related language; I agree that this is inappropriate as well.

34) The Discussion section says "If human classification of these images strongly correlates with DCNNs, as Zhou and Firestone, (2019) observed". But Zhou and Firestone, 2019 do not observe this, for all of the reasons stated above. And this is an especially unfortunate example, since it not only misunderstands Zhou and Firestone, 2019 but also uses a technical term in our field – "correlate", and even "strongly correlate" – that doesn't correspond to anything Zhou and Firestone, 2019 did. Again, Zhou and Firestone, 2019's conclusion is that there is more overlap than would be expected by chance.

---

## [Author Response]

Essential revisions:While the reviewers felt that the paper was of sufficient interest for publication (and a fraction were very enthusiastic about its contributions to the literature and its eventual acceptance at eLife), there were five key points that would be essential to address before the article could be accepted, enumerated below. Additionally, we also ask the authors to also address the specific comments in the section below.1) All three reviewers felt strongly that the language/framing of the paper should be tempered to more constructively reflect the relationship of this paper to the literature and to prevent misinterpretation. Specifically, although the discussion about which statistical methods should be computed to support the claims is key to the manuscript and an important point, there are several places (see specific comments for details) where the reader could get the impression that the authors are claiming that Zhou and Firestone, 2019 computed statistical tests incorrectly (rather than taking a different approach than suggested by the authors here) by using language such as ""statistically unsound." Discussion of the question of which tests to compute would be valuable to this field at the interface of ML and psychology, and so we ask the authors to provide clarity to the text accordingly.

We agree that the term “statistically unsound” is inappropriate and we have removed this phrase. We have also changed some other terms that, on reflection, were too strong. In order to be more constructive, we have now motivated the section on ‘Reassessing the level of agreement’ differently and refrained from comparing the percent of agreement measured using the two methods. We now make it clear that the two types of statistical analyses answer different questions: while the method of computing agreement used by [Zhou and Firestone, 2019] may be suitable for assessing whether agreement between humans and DCNNs was statistically above chance, it is degree of agreement liable to be mistaken as a suitable method for determining the *degree of agreement*. We have given examples of why measuring agreement in this manner is inappropriate if the goal is to measure the degree of agreement and why the alternative measure, the mean agreement, overcomes these misinterpretations. We hope this discussion of different methods of assessing agreement will be useful for future research investigating agreement between humans and ML algorithms.

2) In their original paper, Zhou and Firestone, 2019 write "We conclude that human intuition is a more reliable guide to machine (mis)classification than has typically been imagined." We ask the authors to comment upon whether that the authors' re-analyses (which still show significantly above-chance classification in Zhou and Firestone, 2019's experiments; e.g., in the authors' Table 2) do support this modest conclusion, even though those analyses are still consistent with a fairly low overall level of human-DCNN agreement.

We indeed find that the agreement between human and DCNN classification is above chance in many cases and discussed in detail why one may obtain this result. In short, our experiments show that this above-chance agreement may be due to (a) the difference between the experimental design under which humans and DCNNs are tested, and (b) some superficial features, such as colour, present within the adversarial images.

If Zhou and Firestone, 2019 were only testing the claim that all adversarial images are completely uninterpretable, then yes, our findings claim. Rather, they were also claiming that *universal surprisingly reliable* “human and machine classification are *robustly* related” (Abstract), that are consistent with their conclusion. However, we would like to note that ZF were not just making this very modest “human intuition is a *surprisingly reliable* guide to machine (mis)classification” (Abstract), that their results suggest a “*core visual features* surprisingly agreement with the machine’s choices” and that “adversarial examples truly do share with the images they are mistaken for”, emphasis added in all cases).

These claims are important as they directly address the central question ZF want to investigate: “Does the human mind resemble the machine-learning systems that mirror its performance?” (Abstract). It is important to emphasize that ZF claimed that the observed agreement was *not* due to superficial features of the adversarial images. This was the main conclusion of their Experiment 2. So the main theoretical claim of ZF is that the agreement they reported highlights some important “meaningful” similarities between CNNs and humans. We show that the design of Experiment 2 was flawed (they did not use a good measure for selecting foil images as detailed in our article) and the low level agreement was indeed due to superficial features, or due to images that were not adversarial at all. In sum, our findings are inconsistent with their stronger conclusions (e.g., “surprisingly universal agreement”), and challenge their claim of how the agreement comes about. This is important given that a growing number of neuroscientists, psychologists, and computer scientists are claiming that CNNs are the best current model of human vision.

3) Similarly, the reviewers often had difficulty understanding how subjectively the "best" and "worst" adversarial images were chosen for the experiments? Ideally, this would be done by, say, polling a random group of people, would everyone agree (measured statistically) with these rankings? In absence of such polling, though, we ask the authors to more explicitly discuss which set of adversarial images are used where and to discuss more fully how these choices lead to the conclusions they draw.

We have now carried out the experiment suggested by the reviewers, where we first polled a random group of participants for “best” and “worst” adversarial images and then used these images for testing a second group Experiment 2 of participants. The results from this experiment echo our previous results in : we again find that agreement drops from above-chance to below-chance when “best” images are replaced by “worst” even though both set of images are confidently classified by DCNNs. As this experiment is better controlled, we have replaced the experiment reported in the previous version of the manuscript with the new experiment.

We would also like to note here that, in hindsight, the labels “best” and “worst” weren’t the best choice (though they are now appropriate given the pre-study). What we wanted to examine was whether agreement between humans and DCNNs is robust for some categories, irrespective of the adversarial image chosen from that category. So all we wanted was an “alternative” adversarial image, rather than the “worst” one. What we showed (and now replicate) is that the specific adversarial image chosen matters – i.e. the agreement between humans and DCNNs is not robust even for a subset of categories.

4) The authors consider the existence of adversarial images as a strong counterpoint to the possibility that humans and DCNNs categorize images in the same way. An alternative is that, for many cases, humans and DCNNs are categorizing images in similar ways, but that for some edge cases, DCNNs fail, and we have not yet figured out how to build a DCNN that recognizes when it is being fooled. But what does it mean to have the same way of categorizing images? In this paper, the implicit view is that if humans and the DCNN disagree on this suite of particular experiments, then they are working in fundamentally different ways. Under this definition, do all humans categorize images in one particular way? Do all DCNNs, including DCNNs with different architectures? To make their claim, the authors need to set clear criteria by which to judge that image categorization is the same across networks, and show that variability across DCNNs is smaller than the variability between humans and DCNNs.

The reviewers raise two very interesting issues and we respond to them in order:

Are adversarial images edge cases? We don’t believe so. There are several reasons: (i) the adversarial images considered in this study are classified by DCNNs with high confidence amongst 1000 output categories showing that the network makes no distinction between these images and other images within the test set, (ii) there are a large variety of adversarial attacks (see Akhtar and Mian, [2018]) and, indeed, countless adversarial images for a given output class, and (iii) adversarial attacks are not limited to one particular architecture, but pervasive across different manifestations of convolutional networks.

Still, research on adversarial attacks is ongoing and it is possible that as architectures and learning algorithms improve and image databases increase in size, it becomes increasingly difficult to generate adversarial images. Therefore, we have revised our manuscript to make sure that we are not claiming that adversarial images are a *current* fundamental problem for DCNNs but continue to be a problem for *current* architectures. Does poor agreement necessarily mean humans and DCNNs work in fundamentally different ways? It is true that we took the poor human-CNN agreement as evidence that CNN and human object classification are very different, and the point of the reviewer is well taken in this regards. And indeed, we do not wish to suggest that DCNNs necessarily agree with each other on adversarial attacks. In our experience, there are many adversarial attacks on which there is no agreement between DCNNs.

However, it is also the case that many adversarial attacks are frequently transferable. In fact, a number of studies are trying to investigate why and under what conditions adversarial attacks transfer, see Goodfellow et al., [2014], Tram`er et al., [2017], Demontis et al., [2019]. So, as the reviewers suggest, a stronger test for judging whether humans and DCNNs are working in fundamentally different ways would be to choose adversarial images that transfer across networks and test human-DCNN agreement on these images. We have now carried out exactly this experiment.

We chose 10 state-of-the-art DCNNs and 20 adversarial images that at least 9 our out 10 network classify with high confidence in the same way (high between-network agreement). In a similar experiment to Experiment 1, we observed that even when network-network agreement is high, human-network agreement remained poor and the degree of agreement between network and humans did not depend on the amount of agreement between Experiment 5 networks. We have added to the manuscript where we report these results.

5) Related, the particular adversarial images to which a network is sensitive are reflective of the training set that determined the DCNN. The DCNN in this study was trained on the ImageNet dataset. How would another artificial network, trained on a different image dataset, classify these adversarial images? This would establish which of their findings are expected of any pair of networks with different training sets.To address 4) and 5), it would be helpful to observe the performance of additional DCNNs, trained on completely separate image banks, on each of the four experiments. For example, in Experiment 4, participants had to distinguish between adversarial images generated for ResNet-18 and AlexNet. For 3), how well does ResNet-18 identify AlexNet adversarial images? For 4) If you had two instantiations of AlexNet ("AlexNet1" and "AlexNet2") but trained them on disjoint sets of images from the same classes, would AlexNet1 agree with AlexNet2 on adversarial images generated for AlexNet2? If it turns out that, for instance, AlexNet1 agrees with AlexNet2 at chance level, then the interpretation of chance-level human performance is not valid.

Again, this is an excellent point that did not occur to us. In response to the related point 4 we have conducted a new Experiment 5 experiment to compare DCNN-DCNN to human-DCNN agreement ( ). The reviewers’ suggestion about comparing networks trained on different datasets is also a good one. There is emerging evidence from machine learning that suggests that many adversarial examples from one training set should transfer to other training sets. See, for example, Goodfellow et al., [2014], who note that “To explain why multiple classifiers assign the same class to adversarial examples, we hypothesize that neural networks trained with current methodologies all resemble the linear classifier learned on the same training set. This reference classifier is able to learn approximately the same classification weights when trained on different subsets of the training set, simply because machine learning algorithms are able to generalize. The stability of the underlying classification weights in turn results in the stability of adversarial examples.”

However, the influence of training sets on adversarial attacks (and DCNN representations, in general) is still an active field of investigation, so we have modified the manuscript to acknowledge that the different visual experiences of humans and DCNNs could be one of the factors that influences the poor human-DCNN agreement (see the penultimate paragraph in the Discussion section). In which case, improving the training of DCNNs (and perhaps additionally modifying their architectures) may lead to higher agreement and DCNNs that provide a better theory of the ventral visual pathway.

1) DMB's summary of Zhou and Firestone, 2019's result in the Introduction is ambiguous and open to misreading. Readers might hear DMB's paragraph as stating that Zhou and Firestone, 2019 reported that humans agreed with machines 90% of the time they were asked, which of course Zhou and Firestone, 2019 do not report. A more precise and accurate summary of Zhou and Firestone, 2019's result would be something like this:2). "The authors took a range of published adversarial images that were claimed to be uninterpretable by humans, and in a series of experiments, they showed those images to human subjects next to the DCNN's preferred label and various foil labels. They reported that, over the course of an experimental session, a high percentage of participants (often close to 90%) chose the DCNN's preferred label more often than would be expected by chance."

We have reworded the section to remove any ambiguity. The new wording is “…in a series of experiments, they showed those images to human subjects next to the DCNN’s preferred label and various foil labels. They reported that, over the course of an experimental session, a high percentage of participants (often close to 90%) chose the DCNN’s preferred label at above-chance rates.” Please note that we have not used the phrase “than would be expected by chance” suggested by the reviewers, but instead “at above-chance rates” for two reasons: firstly, this phrase is an exact quote from Zhou and Firestone, 2019, where they state, “98% of observers chose the machine’s label at abovechance rates”, and secondly, because the 90% figure does not refer to percentage of participants who responded more often than expected by chance as ZF counted participants who agreed exactly at chance levels as well as participants who did not agree significantly above chance (please see subsection ‘Reassessing the level of agreement in Zhou and Firestone, 2019’).

3). The final sentence of the Abstract needs to be tempered somewhat. DCNNs are a "promising model," in particular for many aspects of lower-level vision, but they are not a complete model.

We have now revised the final sentence of the abstract to “We conclude that adversarial images still pose a challenge to theorists using DCNNs as models of human vision.”

4) Introduction: Correct the spelling of inferotemporal cortex.

Done.

5) Introduction: The paper revolves on the analysis of the classification of adversarial images, yet beyond showing examples of types of adversarial image, there is not a clear definition of what an adversarial image is and to what extent adversarial images are classified into types (e.g., 'fooling' or 'naturalistic'). Are these all the types of adversarial images?

We have now provided a definition and a reference for adversarial attacks in the Introduction. There is no formal classification into types, we use ’fooling’ as the term introduced by Nguyen et al., [2015] for images which contain no objects but are confidently classified as a specific class. Naturalistic is a term we use to show adversarial attacks which do contain objects seen in the real world but are perturbed in some way to become adversarial.

6) Subsection “Reassessing the level of agreement in Zhou and Firestone (2019)” and Materials and methods section: Why are choices are independent in the null model? Was the task constructed so that choices may repeat? If so, then this is fine. On the other hand, if only 48 images and 48 labels are presented, a participant may treat this as a 'matching' task inducing correlations between choices. For 48 items, the correction to the null distribution of number of correct choices is minor, but it would not be for a smaller number of items. Either way, a clearer description of the task design is warranted.

Indeed, the task is designed in such a way that the trials are independent and the participant can independently choose a label (amongst 48) on each trial. In order to clarify this, we have added the following line: “Each trial is independent, so a participant can choose any of the 48 labels for each image”.

7) Subsection “Reassessing the level of agreement in Zhou and Firestone (2019)”and Table 1: Please explain what the 'Images' column means in this table. Up to this point, the text has referred only to the participant-level analysis.

We have now removed this Table to avoid problems with incorrect characterization of Zhou and Firestone, 2019’s methods (see response to comment (1) under Essential revisions above).

8) Subsection “Reassessing the basis of the agreement in Zhou and Firestone (2019)”: grammar – missing definite article.

Added.

9) Subsection “Reassessing the basis of the agreement in Zhou and Firestone (2019)”: grammar – colloquial usage.

We are not quite sure what the reviewer meant. Could you kindly clarify?

10) Discussion section: This final conclusion, that DCNNs that also incorporate a number of additional features would be less sensitive to adversarial attacks, is interesting. I would recommend reading and possibly citing work on "capsule networks" from Hinton/DeepMind.

Thanks – we have added a citation to capsule networks when discussing how different architectures may solve some of the current issues (Discussion section).

11) Appendix 1—figure 4: check cross-reference – there's a "?"

Done.

12) Critiques of scientific research, including this one, are important, valuable, and welcome. At the same time, a core obligation of any such critique is to present its target charitably, and to engage its strongest arguments. I'm worried that this paper does not do that in its current form. A key piece of context that, in my opinion, DMB's paper loses sight of is that Zhou and Firestone, 2019 were directly motivated by Nguyen et al.'s statement that their fooling images were "completely" and "totally unrecognizable to human eyes" – emphasis on "completely" and "totally". Zhou and Firestone, 2019 noticed that many of the images did seem to have some features in common with their target class, and that Nguyen et al. had collected no human data on this; and so Zhou and Firestone, 2019 simply set out to show that the fooling images in Nguyen et al., were not, as a rule, "totally unrecognizable". It's my strong opinion, even after reading and accepting much of what DMB say, that Zhou and Firestone, 2019's data and analyses (and even DMB's reanalyses) still support Zhou and Firestone, 2019's modest conclusion, and I strongly encourage the authors to consider the same. Note that even Nguyen himself agreed; he wrote to us after reading Zhou and Firestone, 2019's paper to say: "After reading this paper, I develop a stronger intuition that humans can decipher very well those robust, and model-transferrable AXs" and that "I personally agree that the phrase 'totally unrecognizable to human eyes' might have been an overstatement. Thanks to your work for pointing it out!". I agree with that statement too.Indeed, throughout the paper, DMB write statements like Zhou and Firestone, 2019's "conclusions are not justified" [Introduction], but without actually stating what Zhou and Firestone, 2019's conclusions are. In fact, Zhou and Firestone, 2019's conclusions are clear and explicit:- "We conclude that human intuition is a more reliable guide to machine (mis)classification than has typically been imagined" (final paragraph of Zhou and Firestone, 2019's Introduction, where "than has typically been imagined" refers to Nguyen et al.'s claim).- "This implies at least some meaningful degree of similarity in the image features that humans and machines prioritize – or can prioritize – when associating an image with a label" (first paragraph of Zhou and Firestone, 2019's Discussion).and- "The present results suggest that this particular challenge to notions of human-machine similarity may not be as simple as it appears (though there may of course be other reasons to doubt the similarity of humans and machines)" (first paragraph of Zhou and Firestone, 2019's Discussion section).In other words, Zhou and Firestone, 2019's conclusion is not that there's a high level of agreement (though see more on that below), or that DCNNs are good models of human vision, etc.; it is simply that the images are not "totally unrecognizable" in the way Nguyen et al. had claimed. And there can be no doubt that this is Zhou and Firestone, 2019's conclusion, since the only time Zhou and Firestone, 2019 even use the word "conclude" is in the above quote.It is my strong opinion, perhaps my strongest opinion throughout this review, that DMB have not painted that conclusion in an accurate, fair, or charitable way. And so it is also my very strong opinion that the paper must be revised to state that conclusion accurately, and to consider whether the data as a whole (including Zhou and Firestone, 2019's and DMB's analyses) truly are inconsistent with it. Human-machine agreement can be low (as DMB suggest) while still being consistent with Zhou and Firestone, 2019's conclusion that the images are not "totally unrecognizable". There is not as much disagreement between Zhou and Firestone, 2019 and DMB as DMB makes it seem, and there's no reason not to acknowledge that agreement when it exists.

We are not privy to the authors’ intentions, nor to the personal correspondence with Ahn Nguyen and, like most researchers in the field, only have what’s written in the paper to go by. It is true that Nguyen et al., (2015) had used the term “totally unrecognizable” in their paper. However, they did not make the claim that *all* images produced by their algorithm are totally unrecognizable, rather “It *is possible* to produce images totally unrecognizable to human eyes that DNNs believe with near certainty are familiar objects” (emphasis added). Our investigation shows that this statement still holds. There are many images on which agreement is at chance or below-chance levels. The reviewer claims that the entire intent of Zhou and Firestone, 2019 was to show that some images had some features that were common with the target class. But this observation is already present in Nguyen et al., (2015), who write: “In this paper we focus on the fact that there exist images that DNNs declare with near-certainty to be of a class, but are unrecognizable as such. However, it is also interesting that some generated images are recognizable as members of their target class once the class label is known.”

The motivation underlying our manuscript is simply to investigate the theoretical question: are there important similarities between CNN and human object categorisation? This also seems to be the question that interests Zhou and Firestone, 2019 – the first line of their Abstract reads: “Does the human mind resemble the machine-learning systems that mirror its performance?” When ZF find that the agreement between humans and DCNNs is above chance, presumably this is interesting because they think their results help answer this question – i.e. the above-chance agreement implies that human mind does resemble the DCNN. This is the theoretically important position on which we differ. We believe, our results show that the above-chance agreement can arise even though the two systems fundamentally differ from each other.

Still, it is not our goal or our place to speculate on Zhou and Firestone’s intentions. So we have revised the manuscript, so that it is clear that the motivation underlying our work is the above theoretical question. We have also substantially rewritten the section where we reanalyse results from Zhou and Firestone, 2019. Instead of focusing on how our analysis compares to theirs, we have discussed the relative merits (and goals) of the two analyses.

13) The authors claim that Zhou and Firestone, 2019 reported a "surprisingly large agreement between humans and DCNNs" [Introduction]; "the reported level of agreement is nearly 90%" [Results section]. These statements are inaccurate, or at least imprecise. Zhou and Firestone, 2019 do not report a "level of agreement of nearly 90%": "level of agreement" is a new phrase that DMB use (appearing nowhere in Zhou and Firestone, 2019's paper) that makes it seem as though Zhou and Firestone, 2019 reported something they did not, and that also makes it seem that Zhou and Firestone, 2019's measure is aimed at the same quantity as DMB's measure. Additional clarity here would be appreciated, since this comes up later (e.g., in saying that the agreement was "lower than reported", which I worry is misleading – the two measures simply report different quantities).As noted above we have reworded this point about 90% agreement to avoid any ambiguity. We would like to note here that, even though Zhou and Firestone, 2019 may not have used the term “level of agreement”, they do make statements that imply a large degree of agreement throughout their manuscript. For example, in their abstract, they state that “Human intuition may be a *surprisingly reliable* guide to machine (mis)classification” and in the main text they state “98% of observers chose the machine’s label at above-chance rates, suggesting *surprisingly universal* agreement with the machine’s choices”. In our manuscript (subsection *“Reassessing the level of agreement in Zhou and Firestone (2019)”*), we have clarified why the statistic used by Zhou and Firestone, 2019 should not be used to make such statements.14) Similarly, DMB wait too long to note that Zhou and Firestone, 2019 already computed DMB's "average agreement" measure; indeed, it was the very first analysis in Zhou and Firestone, 2019's paper, but DMB only later acknowledge this, instead starting with Zhou and Firestone, 2019's Experiment 3 two pages earlier. In Experiment 1, Zhou and Firestone, 2019 state: "Classification "accuracy" (i.e., agreement with the machine's classification) was 74%, well above chance accuracy of 50% (95% confidence interval: [72.9, 75.5%]; two-sided binomial probability test: p < 0.001)". Indeed, this alone shows how it is misleading of DMB to claim that Zhou and Firestone, 2019 report a "level of agreement" near ceiling; Zhou and Firestone, 2019 are clear that "agreement with the machine's classification" is 74%, just as DMB report.

Zhou and Firestone, 2019 only report average agreement for Experiment 1, and we make this point clearly. Please also see responses to comments (12) and (13) above.

15) I stated above that I still think Zhou and Firestone, 2019's analyses support their conclusions. To see why, consider an analogy. Suppose someone claimed that dieting is "completely" and "totally" ineffective for losing weight, as Nguyen et al., claimed that their fooling images were "totally unrecognizable to human eyes". If you were skeptical (as Zhou and Firestone, 2019 were), you could assign subjects to a diet and see if they lose weight. There are then two standard approaches to evaluating whether the diet was "totally ineffective". You could (1) ask how much weight the cohort lost as a whole, and whether that amount deviates significantly from what would be expected under the null; or you could (2) ask how many people lost weight, and ask whether that number significantly deviates from what would be expected under the null. For example, you could (1) discover that the cohort lost 10% of their bodyweight on average, and that this number deviated significantly from 0%; or you could (2) discover that 95% of dieters lost some amount of weight (whether that amount was a lot or a little), and that this number significantly deviated from 50%.Zhou and Firestone, 2019 take both approaches – (1) and (2) – in their Experiment 1, but then settle on (2) for the rest, because it seemed better suited to refute the claim that adversarial images are "completely unrecognizable to humans". By contrast, (1) is an approach DMB prefer, as is evident from Table 2. That's fine; as DMB show in that table, it also confirms Zhou and Firestone, 2019's conclusions by showing significantly above-chance average agreement. (I comment on Table 1 below.) What should be clear, however, is that these two approaches are both valid, and indeed have certain strengths and weaknesses over one another. For example, if approach (1) finds that subjects lost 10% of their bodyweight on average, that still might not refute the claim that the diet is ineffective, since it could be that most subjects gained a small amount of weight while a minority of subjects lost a very large amount of weight – a possibility that would make this measure ill-tailored to refuting claims of ineffectiveness (since it causes weight gain in most people). Conversely, if you used approach (2) and found that 95% of people lost some amount of weight, then that wouldn't tell you how much weight the cohort (or any one person) lost; but it is still informative, because if 95% of people lose weight on a diet (whether that amount is 1 pound or 50), then whatever you think of this diet, it's hard to plausibly claim that it's "completely" and "totally" ineffective (even if it might not be very effective overall). And, of course, both approaches combined would be best. But, again, they're both valid: If 98% of subjects pick the machine's label numerically more often than not, and if 98% is significantly higher than 50%, then it just can't be true that these images are "completely" and "totally" undecipherable – even if any given subject is only agreeing a bit, and even if that agreement is not even statistically significant within a single subject (just like one needn't say of any given dieter that they lost a "significant" amount of weight; if 95% of dieters lose weight, then that is telling, as long as 95% significantly differs from 50% in the sample).So DMB are being too critical. There are multiple ways to understand these data, depending on the researchers' goals. DMB are correct to say that Zhou and Firestone, 2019's analyses in Experiment 3 "assign the same importance to a participant that agrees on 2 out of 48 trials as a participant who agrees on all 48 trials with the DCNN" [84]; but this just isn't a problem for Zhou and Firestone, 2019's research question. Zhou and Firestone, 2019's goal was to refute the claim that the fooling images in Nguyen et al. were "completely unrecognizable to humans". Perhaps DMB do not share this goal; but that is no reason to call Zhou and Firestone, 2019's approach "misleading and statistically unsound". Both Zhou and Firestone, 2019's approach and DMB's approach are fine. And, again, both approaches ended up coming out Zhou and Firestone, 2019's way, as DMB confirm in Table 2.The paper should be revised to reflect this. There is not actually a large disagreement here between Zhou and Firestone, 2019 and DMB as regards to how Zhou and Firestone, 2019's data interact with Nguyen et al.'s claim (which is what Zhou and Firestone, 2019 were after). To give an example (though of course DMB could use language other than mine), the paper could say something like "Whereas the analyses Zhou and Firestone, 2019 use may have been sufficient to refute previous claims that such images were "totally unrecognizable to human eyes", those analyses are still consistent with a fairly low overall level of human-machine agreement." (But of course, much else would then have to change too, since DMB are so persistent in calling Zhou and Firestone, 2019's conclusions unjustified.)

We have responded to this point above (comments (2), (12) and (13)) – the reviewer is mischaracterizing the claim of Nguyen et al., as well as the claims made by Zhou and Firestone, 2019.

16) A central issue with the re-analyses is how much DMB's discussion focuses on Zhou and Firestone, 2019's Experiment 3b. In my opinion, this was a strange experiment to choose, especially when it comes to giving a charitable critique. The entire purpose of Zhou and Firestone, 2019's Experiment 3b was to make human-machine agreement as difficult as possible, and this is the single least-discussed experiment (of 8) in Zhou and Firestone, 2019's entire paper – not the cornerstone of Zhou and Firestone, 2019's claims in any way. When Zhou and Firestone, 2019 observed above-chance human-machine agreement in Experiment 1, there was a worry that above-chance agreement when given two alternatives wasn't so impressive. So, Experiments 3a and 3b presented all the labels for every image at once, to see if subjects could show above-chance agreement even in extremely taxing circumstances. Zhou and Firestone, 2019 were very explicit about this: Zhou and Firestone, 2019 say "These results suggest that humans show general agreement with the machine even in the taxing and unnatural circumstance of choosing their classification from dozens of labels displayed simultaneously". So Zhou and Firestone, 2019 already described this task as "taxing and unnatural" and unlikely to produce high human-machine agreement. And indeed it's certain that subjects do not actually read all the labels before making judgments, because they often answer in just a few seconds – not enough time to have looked at all the labels. So, of course agreement will be low – that was the point of that experiment.Indeed it is possible to see here – http://www.czf.perceptionresearch.org/adversarial/expts/texture48.html – how imposing that is, whereas Experiment 1 – here http://www.czf.perceptionresearch.org/adversarial/expts/texture.html – feels much more natural. Yet, subjects in Experiment 3a and 3b still showed a mean agreement of ~10%, rather than the chance level of ~2%. We thought that was impressive, given the circumstances. DMB may disagree, but that is simply a difference in opinion or taste, not an undermining of Zhou and Firestone, 2019's claims. (Indeed, DMB later describe this result as "striking"; strikingly high? If so, why not say so earlier?)To be fairer and more accurate, the relevant claims in DMB's paper should also include Zhou and Firestone, 2019's Experiment 1 as an example. For example, subsection “Reassessing the level of agreement in Zhou and Firestone (2019)” should first describe how Zhou and Firestone, 2019 analyzed Experiment 1, and then describe Experiment 3. Figure 1.1 should include Experiment 1 and Experiment 3. And so on. Otherwise, I worry that DMB might be cherry picking, choosing the examples from Zhou and Firestone, 2019 that they feel are weakest, rather than the ones they feel are strongest. For example, DMB state in subsection “Reassessing the basis of the agreement in Zhou and Firestone (2019)” that "For the rest of the images agreement is at or below chance levels". But that's only true of Experiment 3; for Zhou and Firestone, 2019's Experiment 1, which was not designed to be so taxing on subjects, even DMB's binomial analysis shows that agreement is significantly above chance on over 85% of images. It is potentially misleading (if not simply false) to claim that "For the rest of the images agreement is at or below chance levels". This discussion should be more balanced, and either focus on Experiment 1 or at least include parallel analyses for Experiment 1's results whenever DMB discuss Experiment 3.

The reason for focusing on Experiment 3 was, in fact, exactly because the results there were most impressive. In our view, this is the experiment that comes closest to testing participants under similar conditions to DCNNs: participants see a large set of foils to the target class. Experiment 1 only shows one foil and, as demonstrated by our Experiment 1, choosing a random foil may inflate the degree of agreement. See responses (12) and (13) for other points about reanalysis.

17) Beyond the re-analyses, I'm worried that DMB present their views as alternatives to Zhou and Firestone, 2019, when in fact many of these views were already explicitly articulated by Zhou and Firestone, 2019. I've already emphasized how this is true of Zhou and Firestone, 2019's main conclusion. But another example is DMB's suggestion that human-machine agreement on these images reflects "participants making educated guesses based on some superficial features (such as colour) within images and the limited response alternatives presented to them" [Introduction]. Truly, this is already Zhou and Firestone, 2019's exact view about those experiments. Again, with quotes, Zhou and Firestone, 2019 say that subjects "may have achieved this reliable classification not by discerning any meaningful resemblance between the images and their CNN-generated labels, but instead by identifying very superficial commonalities between them (e.g., preferring "bagel" to "pinwheel" for an orange yellow blob simply because bagels are also orange-yellow in color)". That applies to Experiments 1 and 3, which use the same labels.DMB subtly acknowledge this elsewhere, but otherwise they present this hypothesis as if it is original to them. Consider subsection “Reassessing the basis of the agreement in Zhou and Firestone (2019)” or subsection “Experiment 5: Transferable adversarial images”, which says of Elsayed et al., that "These findings are consistent with our observation that some adversarial images capture some superficial features that can be used by participants to make classification decisions" (emphasis added). But this is already Zhou and Firestone, 2019's exact hypothesis. Both Zhou and Firestone, 2019 and DMB believe subjects are making educated guesses. (Indeed, Zhou and Firestone, 2019 think that may be the right way to describe what the CNN is doing too: Zhou and Firestone, 2019 write "both the CNNs' behavior and the humans' behavior might be readily interpreted as simply playing along with picking whichever label is most appropriate for an image", and "CNNs are.… forced to play the game of picking whichever label in their repertoire best matches an image (as were the humans in our experiments)".Perhaps "consistent with Zhou and Firestone, 2019's and our observation" would be a better statement for this and other examples. But it's simply not right to portray this as DMB's own or original interpretation; it is precisely Zhou and Firestone, 2019's interpretation, as these quotes show. This is another example where DMB portray disagreement that may not really exist, and so should attribute this interpretation to Zhou and Firestone, 2019.

The reviewer’s statement that ZF claimed that agreement was based on making educated guesses based on superficial features is false. The quote mentioned by the reviewer (“may have achieved... superficial commonalities between them”) is taken from the motivation of their Experiment 2, which was designed to test whether agreement was driven by superficial features and the findings were taken to refute this hypothesis. The authors wrote: “Again, human observers agreed with the machine’s classifications: 91% of observers tended to choose the machine’s first choice over its second choice, and 71% of the images showed human-machine agreement (Figure 2D). Evidently, humans can appreciate deeper features within adversarial images that distinguish the CNN’s primary classification from closely competing alternatives.” Thus, the view that participants are making educated guesses based on some superficial features is the exact opposite of the conclusion drawn by Zhou and Firestone, 2019'.

18) "Using a statistic that treats 45% agreement as chance is liable to be misinterpreted by readers" [subsection “Reassessing the level of agreement in Zhou and Firestone (2019)”]. Perhaps this is true; I agree that Zhou and Firestone, 2019 could have been clearer. But doesn't this support the opposite of DMB's point? The reader that DMB are imagining would normally take chance to be 50% (as Zhou and Firestone, 2019's figures show, perhaps unclearly), whereas the true value of above-chance-performing participants is 88% or 90%. So the fact that chance is 45% rather than 50% makes the result more impressive, not less impressive, since 88% is even farther above chance than the reader was led to believe.

What are the possible interpretations of the statement “98% of observers chose the machine’s label at above-chance rates, suggesting *surprisingly universal* agreement with the machine’s choices” (Zhou and Firestone, 2019, emphasis added)? One possible interpretation is that if you ask 100 humans to classify an adversarial image, 98 humans (on average) will choose the same label as the machine. This would indeed be a surprisingly universal agreement with the machine’s choice. However, this interpretation turns out to be false. The crucial bit of the statement lies in the phrase “at above-chance rates” – i.e., 98 out of 100 participants agree with the machine if agreement is evaluated as same choice on 50% or more of trials. In each of these trials, participants were given two choices, one of which was the label chosen by the machine and the other one a random label from ImageNet. Put this way, the results are less surprising. As the reviewer agrees (point 17 above) such levels of agreement can arise out of participants making educated guesses based on superficial features present within these images. Indeed, when various experimental factors are controlled, the agreement is scarcely above chance. We did not want to imply that Zhou and Firestone, 2019 intentionally mischaracterized their results (though perhaps the use of the phrase “surprisingly universal” was unfortunate), but that such a misinterpretation of their results and analysis is possible. Our reanalysis tries to present various facets of the data to minimize such misinterpretation. We have revised the manuscript to make this more clear.

19) In the very next paragraph, DMB make the exact kind of misleading statement that they just charged Zhou and Firestone, 2019 with, and in a way that, I worry, makes this part of the reanalysis potentially misleading. DMB say "Measured in this manner (independent 2-tailed binomial tests with a critical p-value of 0.05 for each participant), the agreement in Experiment 3a between DCNN and participants drops from 88% to 57.76%" [94]. But in fact there is no "drop", because chance changes across these two measures. For the 88% measure, chance is 45%; but for the 57.76% measure, chance is 5% (or even 2.5%), since that's the α value of the test that DMB run. In other words, you would expect 5% of subjects to perform significantly different from chance under the null. So DMB have moved the standard without properly alerting the reader. Indeed, 57.76% when chance is 5% is, if anything, more impressive than 88% when chance is 50%. This paragraph should be revised to reflect this. It could say something like, "Measured in this manner, the agreement in Experiment 3a between DCNN and participants moves from 88% (where chance is 45%) to 57.76% (where chance is 5%)"; it would then be clear that "drops" isn't appropriate. They should also report this for Experiment 1, in line with my comment #3 above.

We have now deleted this paragraph as well as Table 1 and rewritten the Reanalysis section to reflect that the two statistical method allow one to answer different questions.

20) This same issue appears in Table 1. To be more interpretable, that table should not only report these %s but should also report the chance level for each %, just as DMB do for Table 2. That will help the reader understand how to interpret what DMB call the "drop" from higher numbers to lower numbers. I strongly recommend this; the table is, I worry, highly misleading without this correction – i.e., a column for something like "what would be expected by chance" (which, I gather is 5%). Indeed, a problem with the analysis in Table 1 is that, in some sense, there's no "null hypothesis"; there's no standard by which the analysis can decide whether the cohort of images was "totally unrecognizable" or not. The value of Zhou and Firestone, 2019 analysis, and DMB's Table 2, is that it's clear how to reject the null hypothesis: i.e., if significantly more images are numerically above chance than are numerical below chance (Zhou and Firestone, 2019), or if mean agreement is significant above chance (Zhou and Firestone, 2019 and DMB). But DMB's Table 1 analysis doesn't really work that way. So that's why it's crucial to portray what chance is for those tests, in the table itself, so that readers can see that it was 85% agreement when chance was 5%, or 57% when chance was 5%, etc.

We have now removed, what was, Table 1 and instead focused this section on discussing mean agreement as the more appropriate measure for the degree of agreement between humans and DCNNs.

21) A final issue with the reanalyses themselves, and in some ways the biggest one, is just that Zhou and Firestone, 2019's conclusions survive them, in ways that make it extremely confusing to me why DMB draw the conclusions they do. As Table 2 shows, every one of Zhou and Firestone, 2019's experiments continues to show significantly above-chance human-machine agreement, even on the analytical approach that DMB prefer. And DMB acknowledge this: They say "it is nevertheless the case that even these methods show that the overall agreement was above chance" [subsection “Reassessing the basis of the agreement in Zhou and Firestone (2019)”]. First, DMB should not wait this long to say so; they should say as early as possible that their analyses, like ours, show significantly above-chance agreement. But second, this demonstrates that Zhou and Firestone, 2019's conclusions, as stated above, are secure after all. I must repeat again that Zhou and Firestone, 2019's claims, made explicitly in their paper, are as follows: "that human intuition is a more reliable guide to machine (mis)classification than has typically been imagined", and that the results "impl[y] at least some meaningful degree of similarity in the image features that humans and machines prioritize-or can prioritize-when associating an image with a label". DMB confirm that these conclusions are even more robust than Zhou and Firestone, 2019 suggested, since they are supported by Zhou and Firestone, 2019's approach and DMB's.

We indeed find that the mean agreement is above-chance for the experiments carried out by Zhou and Firestone, 2019. We dedicate a substantial portion of the paper discussing why exactly this may be. The experiments we carry out are designed to tease apart these reasons. We show that the above-chance mean agreement is partly due to how the experiments and the stimuli set are chosen and partly due to (some superficial) properties of adversarial images.

22) Moving to the experiments: DMB's abstract states that "it is easy to generate images with no agreement" [16]. I don't see how this claim is justified by their experiments. First, half or more of DMB's experiments reflect a failure to "generate images with no agreement". Experiment 1 shows above-chance agreement: "A single sample t-test comparing the mean agreement level to the fixed value of 25% did show the difference was significant (t(99) = 3.00, p = .0034, d = 0.30)" [179]. Experiment 2 does so as well: when the best and worst cases are combined, their average agreement is above chance – it's only through picking images that DMB think look undecipherable that they were able to find undecipherable images (note that Zhou and Firestone, 2019 never state that all images will be undecipherable, but rather that the procedure for producing such images will tend to produce decipherable images in general; Experiment 2 confirms this). Experiment 3 does so as well: "Participants were slightly above chance for indirectly-encoded MNIST images (t(197) > 6.30, p <.0001, d = 0.44)" [subsection “Experiment 3: Different types of adversarial images”]. (Only Experiment 4 does not; I'll return to that later.)This was confusing. Why, if it is "easy" to generate images with no agreement, did DMB so frequently fail to do so? This conclusion should be altered to something like "While difficult, it is possible to generate images with no agreement"; but surely not "easy"!

The phrase “easy to generate images with no agreement” pertains to Experiment 4. This is the only experiment in which we have generated adversarial images ourselves. All other images were taken from Nguyen et al., (2015). However, we agree that “easy” is an informal expression, so we have replaced this with the more precise phrase: “we find that there are well-known methods of generating adversarial images where humans show no agreement with DCNNs.”

23) Experiment 1 is hard to interpret as run; or, if it is, it doesn't show a lack of human-machine agreement. The authors chose labels that they subjectively felt were good competitors for the DCNN's label, and found that agreement dropped but was still significantly above chance. This is unsurprising and no threat to Zhou and Firestone, 2019's view, for at least three reasons.First, this conclusion was already explicitly reached by Zhou and Firestone, 2019, who also considered this and whose Experiment 2 showed that human-machine agreement drops with more competitive labels. DMB mention Zhou and Firestone, 2019's Experiment 2, and criticize it on other grounds; but those grounds are independent of this aspect of Zhou and Firestone, 2019's discussion. In other words, even granting DMB's criticism of Zhou and Firestone, 2019's Experiment 2, Zhou and Firestone, 2019 still perfectly anticipate the conclusion of DMB's Experiment 1. But again, as above, DMB do not credit Zhou and Firestone, 2019 with this, and instead present this as though it is original to DMB. DMB should acknowledge that their Experiment 1 confirms the results of Zhou and Firestone, 2019's Experiment 2 – that more competitive labels should reduce agreement.Second, DMB's version (but not Zhou and Firestone, 2019's version) is much more difficult to interpret because it is almost a form of "double-dipping": The researchers, DMB, are human beings with visual systems who can appreciate which images look least like certain labels; so they picked the foil labels they thought fit best, and then discovered that other humans (their subjects) agreed. But Zhou and Firestone, 2019 never claimed that humans would pick the DCNN's label as the literal #1 label among all 1000! Again, Zhou and Firestone, 2019's claim is only that humans favor the DCNN's label better than would be expected if the images were "totally unrecognizable to human eyes". So, this experiment is perfectly consistent with Zhou and Firestone, 2019's view. I keep returning to this because DMB have made Zhou and Firestone, 2019's paper the cornerstone of their own paper. If, instead, they just presented four new interesting experiments, they could interpret those experiments in their chosen way, and readers could decide to be persuaded or not. But instead, DMB present these experiments as refuting Zhou and Firestone, 2019. In that case, they have to get Zhou and Firestone, 2019's claims right. All Zhou and Firestone, 2019 need is for the humans to think the DCNN's label fits better than previously thought; Zhou and Firestone, 2019 don't need it to be the very best label.Third, and most crucially, DMB's Experiment 1 still shows above-chance agreement! So it seemingly shows the opposite of DMB's claim (that their experiments show it is easy to generate images with "no agreement"), and continues to support Zhou and Firestone, 2019.

The goal of Experiment 1 was not to discredit Zhou and Firestone but to understand whether humans will agree with DCNN classification if they made their decision under the same conditions as the DCNN. Obviously, it’s impractical to run an experiment where humans have to choose amongst 1000 labels. Therefore, Zhou and Firestone chose the alternative labels randomly. Choosing alternative labels in this fashion is understandable. However, the results from Experiment 1 shows that, had the experiment with 1000 alternative labels been feasible, it would have shown much lower level of agreement as responses would have been distributed over these competing labels. Zhou and Firestone may not be interested in this question, but we feel that it is an important question to address when considering whether there are meaningful similarities in human and DCNN object recognition.

The reviewer is however entirely correct on the dangers of double-dipping in this experiment. To mitigate this effect, we chose alternative competitive labels so that they were not semantically related to each other (see Appendix 2—figure 1). Even so, we agree that even though we find that agreement is close to chance in Experiment 1, this does not necessarily mean that it will be close to chance when the participants see all 1000 categories. Therefore, we do not make this claim in our manuscript. The critical finding of this experiment is that agreement drops considerably when labels are not randomly selected, which shows that participants cannot clearly identify a single category from these images that DCNNs classify with 99% confidence.

24) Experiment 2 is also hard to interpret, for the same reason as Experiment 1, and for an additional reason. It shows that some images are easy to decipher (showing above chance classification), but some are hard to decipher (showing below chance classification). Note that Zhou and Firestone, 2019 explicitly predict this, but again DMB fail to acknowledge this. Zhou and Firestone, 2019 state: "A small minority of the images in the present experiments.… had CNN-generated labels that were actively rejected by human subjects, who failed to pick the CNN's chosen label even compared to a random label drawn from the image set. Such images better meet the ideal of an adversarial example, since the human subject actively rejects the CNN's label". So it is no surprise that it is possible to find images of the "worst-case" type by having a human pick them out; Zhou and Firestone, 2019 already said that should happen, when they wrote "An important question for future work will be whether adversarial attacks can ever be refined to produce only those images that humans cannot decipher, or whether such attacks will always output a mix of human-classifiable and human-unclassifiable images; it may well be that human validation will always be required to produce such truly adversarial images". DMB's Experiment 2 is perfect evidence for Zhou and Firestone, 2019's prediction; they show, just as Zhou and Firestone, 2019 say, that a process that tends to generate decipherable images will also generate some undecipherable ones that a researcher could pick out. This is yet another example of DMB making a claim as if it is original to them, rather than crediting Zhou and Firestone, 2019 with that claim and noting that DMB's results confirm Zhou and Firestone, 2019's claims or predictions. It continues to confuse me why DMB frame things this way. There is no need to write as though there is a disagreement here.

Again, the point of Experiment 2 is not to discredit ZF but to examine the following important question: do humans and CNNs consistently agree with each other on a subset of image categories or only on a subset of images. E.g., do humans and DCNNs agree on what a ‘Tile roof’ looks like. The adversarial image we used in Experiment 1 (random labels condition) for Tile roof showed an agreement of 75% between participants and the DCNN. So, in Experiment 2 we chose a different adversarial image for the same category. In this case, we found that the agreement dropped to ∼17% (i.e. below chance) showing that the agreement wasn’t consistent for this category. This same pattern is reproduced for many categories and overall agreement for alternative images is below chance.

We think that this is informative, and incidentally, contradicts Zhou and Firestone’s claim that human intuition is a *reliable* guide to machine (mis)classification.

25) Indeed, a relevant difference between Zhou and Firestone, 2019's experiments and DMB's Experiment 2 is that Zhou and Firestone, 2019 used images that they didn't even "choose"; they simply used the ones Nguyen et al. displayed in their paper. DMB show that, if the researcher selects a subset of those images with the intent to pick the undecipherable ones, then it is possible to do so. But that's not what's at issue; what's at issue is whether the algorithmic process that generates such images tends to produce only undecipherable images, or a mix of decipherable and undecipherable images. Zhou and Firestone, 2019 chose their images in an unbiased/random way (at least, relying on Nguyen et al.,'s presentation), and found decipherability. That's a crucial difference: "unbiased" vs "biased" selection of images.

Again, the question is not whether DMB’s choice or Zhou and Firestone’s choice (or Nguyen’s choice, for that matter) is the better one, but whether the choice on certain categories matters. Clearly it does. This is problematic if one infers the agreement on certain categories to mean that humans and DCNNs share representations for some categories. Also, we do not claim that all adversarial images are undecipherable. On the contrary, we argue that many adversarial images are clearly interpretable (subsection “Reassessing the basis of the agreement in Zhou and Firestone (2019)”).

26) Can the authors make their experiment code available? I apologize if I missed it, but I only saw the data and images in their OSF archive.

We have uploaded all materials (stimuli) as well as data collected onto OSF. The reason for not uploading the code is that the experiments were conducted in PsyToolkit and PsychoPy (through Pavlovia) which have both undergone several version updates, which would render the code unworkable. We have therefore provided detailed descriptions of the experimental procedure in the Methods section which, along with the uploaded Materials and methods section, should be sufficient for easy replication. Should the reviewer or any reader need the code that we used, we will be happy to provide this upon request.

27) Experiment 3 also contracts DMB's stated conclusions, and confirms Zhou and Firestone, 2019's hypothesis yet again; even though "To our [DMB's] eyes, these MNIST adversarial images looked completely uninterpretable" [subsection “Experiment 3: Different types of adversarial images”], they still were interpretable! As DMB state, "Participants were slightly above chance for indirectly-encoded MNIST images (t(197) > 6.30, p <.0001, d = 0.44)" [subsection “Experiment 3: Different types of adversarial images”]. Indeed, DMB later contradict this result by saying "Experiment 3 showed that it is straightforward to obtain overall chance level performance on the MNIST images"; but that is not what happened, as subsection “Experiment 3: Different types of adversarial images” shows. It was not straightforward; the images, as a group, were deciphered above chance.

There is no contradiction in our claims and conclusions, and Experiment 3 provides no support for Zhou and Firestone, 2019 claims. We found 13.53% agreement for indirect and 10.43% agreement for directly generated images (when chance was 10%). Appendix 2—figure 3(C) illustrates the true nature of the “interpretable” indirectly encoded MNIST results. There was a tendency for participants to choose the digit ’0’ (it was the most frequent choice for 6/20 images and made up a total of 16.8% of responses) and digit ’1’ (a total of 14.26%). Looking at 16/20 stimuli, accounting for that preference, the average agreement falls to exact chance (10.2%). Looking at all the stimuli reveals quite large variability which also negates any claim of general interpretability of these images. The results with the direct and indirect encoded MNIST images clearly challenge the claim that humans have meaningful insights into how DCNNs classify these images. We believe that simply pointing out significance without judging variability, systematic but unrelated tendencies, and effect sizes is not informative enough.

28) Another issue with Experiment 3 is that the study used some labels that would be highly unfamiliar to subjects (or, at least, it seems to have done so; again, the experiment code would be helpful). For example, Appendix 2—figure 3 highlights that subjects were reluctant to call a certain image a "Lesser Panda". The authors seem to interpret this as meaning that subjects believed the image did not look like a Lesser Panda. But, of course, an alternative is that the subjects don't know what a Lesser Panda is. Isn't this an alternative explanation? If so, it would have nothing to do with decipherability, and instead to do with whether subjects know what certain ImageNet labels refer to. Indeed, ImageNet gives "Red Panda" as an alternative, but DMB chose to use "Lesser Panda"; why? The image contains a central red patch; I'd strongly predict that subjects would have classified it above chance if DMB hadn't chosen the much more obscure label "Lesser Panda".

We don’t think the obscurity of the label is an explanation for why we observe a chance-level or below-chance agreement on some of the images. Firstly, the labels for other images in this experiment where we observe ator below-chance performance are ‘cheetah’, ‘golden retriever’, ‘stopwatch’ and ‘soccer ball’. There is no reason to suspect that participants don’t know what these categories look like any more so than categories that show above-chance agreement. (All these category labels are already available along with the images uploaded on OSF). Secondly, even if participants don’t know what a ‘Lesser Panda’ is, surely they know what a Panda is – that is enough information to distinguish it from alternative labels, such as ‘centipede’, ‘stopwatch’, ‘cheetah’, etc.

Moreover, if the reviewer is correct and agreement increases on swapping the label from ‘Lesser Panda’ to ‘Red Panda’ because the image contains a central red patch, this is entirely in line with our argument in the manuscript: participants choose labels by making educated guesses based on superficial features (such as colour) present within these images. It also directly contradicts the conclusion of Experiment 2 in ZF, where they tested whether agreement was due to “superficial commonalities” and found instead that “humans can appreciate deeper features within adversarial images that distinguish the CNN’s primary classification from closely competing alternatives” (Zhou and Firestone, 2019').

29) Experiment 4 is the most interesting contribution of the paper. Indeed, considering everything I have written above – which I acknowledge has been quite negative – Experiment 4 seems interpretable and really does show chance-level classification. This is the part of the paper that could make a new and meaningful contribution, in way that is not misleading, does not misconstrue Zhou and Firestone, 2019's conclusions, and does not show above-chance deciphering. Zhou and Firestone, 2019 do have a reply to this – it is a version of the "acuity" reply, which DMB consider for their indirectly encoded images (and rightly reject), but not for their directly encoded ones. DMB cite evidence that suggests this: Elsayed et al., showed that when properties of the primate retina are incorporated into a CNN that is adversarially attacked, those attacks do look quite decipherable to humans. But the fact that we would give this reply doesn't really bear on this review of DMB's paper, or its publishability. So even though I disagree with DMB's interpretation of Experiment 4, I have no problem with it in the way I do with the rest of the paper.

We are pleased the reviewer likes this experiment.

30) DMB's item-level analysis was described in a way I found confusing or maybe even misleading. Consider subsection “Reassessing the basis of the agreement in Zhou and Firestone (2019)”: "agreement on many images (21∕48) was at or below chance levels. This indicates that the agreement is largely driven by a subset of adversarial images". But what DMB call a "subset" was in fact a majority of images! 27/48 here, and 41/48 in Experiment 1. (Indeed, it's not made explicit where this number comes from; in Experiment 3, 39/48 are numerically above chance and 9/48 are numerically below chance. The authors should clarify when they are referring to numerically above chance and when significantly above chance, and to always report what chance is for these analyses.) Again, I don't mean to say this disrespectfully, but it frequently feels that DMB are going out of their way to describe Zhou and Firestone, 2019's results in uncharitable ways. DMB note that 85% of images in Experiment 1 are significantly agreed-with above chance, and that 57% are significantly above chance in Experiment 3b. (And if they combined the data from 3a and 3b, which they do elsewhere, they would find that this total is closer to 70%). Calling a majority of images (85%, 70%, or 57%) a "subset" is of course literally true, but it implies that it's somehow a small number of images, when it fact it's most of the images! Especially when chance for all of these statistics is only 5%. Please refer to it that way, rather than imply it's some kind of small number.

27/48 is a subset, and we provide the numbers in the relevant text. We don’t see how any of this is misleading.

30) Please also annotate the lines in Appendix 1—figure 1, with labels, so that it is immediately clear to the reader that the black line is chance.

Done.

31) The authors should always make clear when the stimuli they show to readers were chosen algorithmically or chosen by the authors' own subjective impression of them. For example, Figure 3 could give the impression that authors have some procedure to generate best-case and worst-case images; indeed, I originally interpreted the figure that way. But in fact, I now understand that this just reflects their own choices about which images look least like their target class. So this caption should say so – something like, "Example of best-case and worst-case images for the same category ('penguin'), as judged by the present authors, and as used in Experiment 2". And so on elsewhere, including the generation of labels. Another example is Appendix 2—figure 2, which says "these were judged to contain the least number of features in common with the target category". It would be clearer to say "we judged these images to contain the least…". I know this does happen in some places, but even there it is confusing (for example, DMB say they picked images from "each category"; but in fact I believe it's just each image from one of 10 categories, right? It's worth being especially clear here on both counts).

Please see response (3) to Essential revisions above.

32) Similarly, in subsection “Experiment 1: Response alternatives” ("We chose a subset of ten images from the 48 that were used by Zhou and Firestone, 2019 and identified four competitive response alternatives (from amongst the 1000 categories in ImageNet) for each of these images"), the authors should state the procedure they used to do this. Why did they choose only 10 images? How did they choose the response alternatives? They give some examples of their negative criteria (e.g., no semantic overlap), but that still leaves out a lot of the selection process. Of course, if the answer is just that they selected in advance the images and labels that they thought would show low agreement, it's important to say so explicitly (though that would, of course, undermine aspects of the experiment's interpretation). I'm also confused why DMB excluded "basketball" from the "baseball" image; they are both kinds of balls, but of course people would be unlikely to visually confuse them – isn't this a (likely unintentionally) self-serving experimental decision?

The number of images (10) was decided upon to match Experiment 3, which includes MNIST stimuli which has 10 categories. Since that experiment had a 2x2 design we wanted all conditions to have the same number of stimuli. Consequently, the same categories were used in both experiments. The images were selected at random from the 48 used by Zhou and Firestone (Experiment 3b) and then checked for whether or not they contain images like ’chainlink fence’ which trivially look like interpretable images from the category. Additionally, we checked the average agreement as computed by Zhou and Firestone to make sure the selection not biased towards low agreement as computed by Zhou and Firestone. In fact, 9 out of 10 images chosen show above chance agreement as computed by Zhou and Firestone (90% compared to 81.25% of the overall set from their experiment). The average agreement being 7.87% which was not significantly different to the average for the entire 48 image set (9.96%). The label choice for the competitive label condition excluded other objects from the categories closely related to the target category exactly in order not to bias results in favour of the hypothesis that competitive labels would decrease agreement levels (e.g. not choosing ’acoustic guitar’ as the foil for ’electric guitar’).

33) I may well have an overly sensitive ear here, but there is a feeling throughout the paper that DMB believe Zhou and Firestone, 2019 behaved in a sneaky or obfuscatory way in reporting their results. Multiple colleagues have shared with me a similar reading of DMB's paper (after seeing their publicly posted preprint), wondering why it is so sharply worded and insinuatory. I have to say I agree. This is especially unfortunate because nothing could be farther from the truth: Zhou and Firestone, 2019 were transparent about all of these analyses, and just in case we weren't, we proactively made all of our data publicly available so that researchers could know exactly what we did – that, of course, is how DMB acquired the data in the first place. (DMB do not mention this either; it would be informative to the reader, and perhaps more collegial of DMB, to state that the reason they were able to reanalyze Zhou and Firestone, 2019's data was because of Zhou and Firestone, 2019's proactive transparency in making them public.) I hope that a revision, whether here or elsewhere, can be more respectful of other researchers' motivations and not insinuate hidden analyses or selective reporting.For me, the tone is present throughout, such that it is hard to point out every example. Here are some:- "Our first step, in trying to understand the surprisingly large agreement between humans and DCNNs observed by Zhou and Firestone, 2019, was to reassess how they measured this agreement" [subsection “Reassessing the level of agreement in Zhou and Firestone (2019)”]. But there was no need for DMB to find themselves "trying to understand" these analyses, as if those analyses were somehow obscure or hidden; the analyses, code, and data were made publicly available alongside the paper itself.- "We noticed that Z and F used images designed to fool DCNNs trained on images from ImageNet, but did not consider the adversarial images designed to fool a network trained on MNIST dataset" [subsection “Experiment 3: Different types of adversarial images”]. Again, DMB write as if they are suspicious or something. But the reasons are simply that (a) Nguyen et al., highlight their ImageNet images much more (e.g., in their Figure 1), and (b) MNIST-trained networks aren't usually claimed to resemble human vision in the same way as ImageNet-trained networks. Moreover, Zhou and Firestone, 2019 do "consider the adversarial images designed to fool a network trained on MNIST dataset"; that's Zhou and Firestone, 2019's Experiment 5. So this language is not only unnecessary, but also even false.- "when we examined their results more carefully, the level of agreement was much lower than reported" [Discussion section]. There are two problems here. First, what does "more carefully" mean in this context? More carefully than Zhou and Firestone, 2019? That really seems to imply that Zhou and Firestone, 2019 made an error or something, which DMB do not in fact believe as far as I know. DMB simply prefer another measure, not a "more careful" one, and as DMB acknowledge, Zhou and Firestone, 2019 already carry out some of their preferred analyses. And "much lower than reported" is simply inaccurate; it's fine that DMB prefer a different measure, but that's not the same as Zhou and Firestone, 2019 falsely or inaccurately reporting theirs. All instances of "lower than reported" simply must be revised; Zhou and Firestone, 2019 reported everything accurately – DMB just prefer a different analysis.- The editor and other reviewers have also flagged "statistically unsound" and related language; I agree that this is inappropriate as well.

We have changed some words that the reviewer objects to (“statistically unsound”), but we do not understand the reaction. None of them (apart from “statistically unsound”) seem to us inappropriate nor suggest that Zhou and Firestone, 2019 behaved in a sneaky or obfuscatory way. To address reviewer 1’s concern we have also changed the sentence “when we examined their results more carefully” to “when we examined their results in more detail”.

34) The Discussion section says "If human classification of these images strongly correlates with DCNNs, as Zhou and Firestone, (2019) observed". But Zhou and Firestone, 2019 do not observe this, for all of the reasons stated above. And this is an especially unfortunate example, since it not only misunderstands Zhou and Firestone, 2019 but also uses a technical term in our field – "correlate", and even "strongly correlate" – that doesn't correspond to anything Zhou and Firestone, 2019 did. Again, Zhou and Firestone, 2019's conclusion is that there is more overlap than would be expected by chance.

Please see our responses (12) and (13) above and (2) in Essential revisions.